

# Combined impacts of climate and land-use change on future water resources in Africa

Celray James Chawanda[1], Albert Nkwasa[1], Wim Thiery[1], Ann van Griensven[1,2]

[1]Department of Hydrology and Hydraulic Engineering, Vrije Universiteit Brussel, 1050 Brussels, Belgium.
[2]IHE-Delft Institute for Water Education

*Correspondence to*: Celray James CHAWANDA (celray.chawanda@vub.be)

**Abstract**. Africa depends on its water resources for hydroelectricity, inland fisheries, and water supply for domestic, industrial, and agricultural operations. Anthropogenic climate change (CC) has changed the state of these water resources. Land use and land cover has also undergone significant changes due to the need to provide resources to a growing population. Yet, the
impact of the Land Use and Land Cover Change (LULCC) in addition to CC on the water resources of Africa is underexplored. Here we investigate how precipitation, evapotranspiration (ET), and river-flow respond to both CC and LULCC scenarios across the entire African continent. We set up a SWAT+ model for Africa and calibrated it using the Hydrological Mass Balance calibration (HMBC) methodology detailed in Chawanda et al., (2020a). The model was subsequently driven by an ensemble of bias-adjusted global climate models to simulate the hydrological cycle under a range of CC and LULCC scenarios.
The results indicate that the Zambezi and the Congo River basins are likely to experience reduced river flows under CC by up to 7% decrease, while the Limpopo will likely have higher river flows. The Niger River basin is likely to experience the largest decrease in river flows in all of Africa due to CC. The Congo River basin has the largest difference in river flows between scenarios with (over 18% increase) and without LULCC (over 20% decrease). The projected changes have implications on agriculture and energy sectors and hence the livelihood of people on the continent. Our results highlight the need to adopt
policies to halt global greenhouse gas emissions and to combat the current trend of deforestation to avoid the high combined impact of CC and LULCC on water resources in Africa.

## 1   Introduction

Africa benefits from surface water in various ways. Surface water resources drive hydroelectricity, inland fisheries, and water supply for domestic, industrial, and agricultural operations. With agriculture as the mainstay of most African economies
(Evans, 2018), irrigation is of importance for local food security, with sustainable irrigation practices having the potential to boost African agricultural productivity in many areas without adverse environmental impacts on freshwater resources (Rosa et al., 2020). Surface water provides most of the irrigation water used in Africa (Frenken, 2005).

Anthropogenic climate change (CC) has resulted in changes in the global state of water resources(Grant et al., 2021; Gudmundsson et al., 2021; Marvel et al., 2018; Padrón et al., 2020; Vanderkelen et al., 2020) . Regarding Africa, Maidment



*et al.* (2015) concluded that there had been a general increase in precipitation from 1983 to 2014 over the Sahel (29 – 43 mm yr[-1] per decade), Southern Africa (12 – 41 mm yr[-1] per decade) while East Africa has dried at -14 – -65 mm yr[-1] per decade in March-May. There is also ample evidence that the African water resources have responded to observed warming through reduced snow and ice cover, increases and decreases in runoff, and changing precipitation patterns, including extremes (IPCC, 2014, 2021). The World Meteorological Organisation (2019) has documented the increase in extreme events attributed to CC,

including the recent shift from arid conditions to heavy rains and floods in 2019 in the Greater Horn of Africa. Further changes in the hydrological cycle due to CC are expected towards the future (Bais et al., 2015; Souverijns et al., 2016; Vanderkelen et al., 2018).

Changes in rainfall patterns linked to natural variability and CC can have profound societal consequences in Africa, where rainfall is crucial for sustaining livelihoods and economic development (Maidment et al., 2015). For example, the 2014 – 2015

drought in Zimbabwe and the 2016 drought in Kenya left millions of people with food shortage (FEWS NET, 2016; Uhe et al., 2018), while crop fields were destroyed and hundreds of thousands of people were displaced in parts of Malawi and Mozambique due to floods triggered by heavy rains in 2014-2015 (Liberto, 2015). The food supply problems posed by climate changes are expected to worsen (Hurlbert et al., 2022; Nkrumah, 2018; Zommers et al., 2020). The Intergovernmental Panel on Climate Change (IPCC, 2014, 2021) pointed out that projected warming in Africa is larger than the global annual mean

warming, and further changes of rainfall patterns are expected. Thus, projecting how the hydrological resources of Africa may respond to CC is of importance for reducing potential adverse impacts of CC and enabling sustainable development.

Land cover has also undergone significant changes driven by the need to provide food and other resources for a growing population (Foley et al., 2005; Lawrence et al., 2016). Between the years 1990 to 1997, 310,000 ha year[-1] of forests were converted to agricultural land (Achard et al., 2002). An update by Hansen et al. (2013) revealed that tropical forests are being

lost at the rate of 53600 ha year[-1], and since the year 2000, Africa has lost millions of hectares of tree cover (Global Forest Watch, n.d.). The loss of forests is expected to accelerate with projected population growth. In addition, Africa is one of the fastest urbanising continents (Beniston et al., 2011; Cartwright, 2015; Ruhiiga, 2013), which is also cutting into forest lands. These Land Use and Land Cover Changes (LULCC) influence the water resources on the continent. Deforestation and urbanisation in Africa have been associated with decreased rainfall and enhanced runoff, increasing the risk of flooding

(Akkermans et al., 2014; Shi et al., 2007). Likewise, irrigation and conservation agriculture may alter local and remote precipitation patterns (De Hertog et al., 2022; Hauser et al., 2019; Hirsch et al., 2018; Thiery et al., 2017, 2020). In West Africa, the loss of forests has contributed to reduced rainfall (Garcia-Carreras & Parker, 2011). The Upper Blue Nile and parts of South Africa have experienced increased surface runoff due to expansion of cultivated land and urbanisation (Gyamfi et al., 2016; Woldesenbet et al., 2017). Thus, LULCC, in addition to CC, poses a threat to water resources availability in Africa.

Only few studies investigated how the hydrology will responded to LULCC in Africa and even fewer have looked at combined impacts of CC and LULCC, and most of these studies have been conducted at small scales (e.g. Mango et al. (2011), Gyamfi



et al. (2016), Warburton et al. (2012)). Thus, the impacts of future LULCC on the hydrology of Africa are underexplored. Nevertheless, such studies are needed to plan for mitigating against potential negative impacts of LULCC on African water resources.

Climate and hydrological models are used to generate future scenarios and quantify water resources at various spatial and temporal scales (Kim et al., 2008). Hydrological models at small spatial scales are often calibrated and validated and are therefore arguably more reliable in their projections (Krysanova et al., 2018; Trambauer et al., 2013). In contrast, global and continental-scale models are often not calibrated but allow for large-scale CC impact assessments (Boulange et al., 2021; Hattermann et al., 2017; Reinecke et al., 2021; Sood & Smakhtin, 2015; Sterl et al., 2020; Tabari et al., 2021; Thiery et al.,
2021; Zaherpour et al., 2018). Chawanda et al. (2020a) and Krysanova et al. (2020) demonstrated that calibration of large-scale models has a substantial impact on the model's projections.

One of the reasons that calibration of large-scale models is currently limited is the large computational requirements of such a procedure. To overcome this limitation, Chawanda et al. (2020a) suggest Hydrological Mass Balance Calibration (HMBC), with less computational and time demands. HMBC uses long-term annual average components of the water balance to calibrate
the model.

This study projects how runoff and river flow across Africa may change under different future CC and LULCC scenarios. First, we use the HMBC methodology (Chawanda et al., 2020a) to calibrate a continental-scale hydrological model and subsequently drive it with an ensemble of bias-adjusted global climate models to simulate future runoff, evapotranspiration (ET) and river flow projections under CC and LULCC scenarios for RCP 2.6, 6.0 and 8.5. Finally, we analyse the changes in
river flows and ET changes in the future under both CC and LULCC scenarios compared to a reference historical period.

## 2    Materials and methods

### 2.1    SWAT+ model

The SWAT model is a process-based hydrological model that is usually applied at a catchment scale. It is a semi-distributed and time-continuous model (Arnold et al., 2012). SWAT+ is a restructured version of SWAT (Arnold et al., 2018; Bieger et
al., 2017). A more detailed description of the SWAT+ model can be found in Chawanda et al., (2020b; Section 2).

### 2.2    Source code adaptations to run land-use change scenarios

By default, the SWAT+ model does not simulate transient land use. We modified the source code to read updated areas for each land use category at the beginning of each simulation year. We achieved this by creating yearly HRU connection files (hru.con) with the naming scheme *hru_xxxx.con* where xxxx is the year for the connection file. We also created yearly files
for the Landscape Unit Elements file (ls_unit.ele) with the analogous naming scheme *ls_unit_xxxx.ele*. These files are used to



update the area of the HRUs and the percent area of LSU they are contained in. If the connection file for a given year does not exist, the previously used HRU connection file is maintained to avoid crashes in model runs where transient land use is not simulated. The same mechanism was also implemented for the landscape unit elements file.

## 2.3 Model setup and calibration

The SWAT+ model setup was done using SWAT+ AW (Chawanda et al., 2020b). The methodology and datasets for model setup and evaluation is described in detail in Chawanda et al. (2020a). The model was set up for the entire African continent, including Madagascar but excluding smaller islands.

Model calibration was done using HMBC using the same simulation period discussed in Section 2.3 (1979 to 1986). The forcing used for calibration runs was from EWEMBI (Lange, 2016). HMBC uses soft data (discussed in Section 1) to calibrate

the model against long-term annual average water balance components. The advantage of HMBC is that it only needs a few simulations to complete the calibration and demands less detailed data making it applicable in calibrating models at continental and global scales. More details on HMBC are discussed in Chawanda et al. (2020a).

Unlike for a previous Southern Africa SWAT+ model application (Chawanda et al., 2020a), the Nile and Congo River basins had very few gauging stations from which long term average surface runoff could be derived. As such, only ET was calibrated

by HMBC in some calibration zones from these river basins. If there is a gauging station downstream, HMBC is applied to calibration zones that have no surface runoff ratio so that they collectively yield the estimated surface runoff ratio downstream. Thus, although HMBC can be used in data scarce areas. It is limited by absence of long-term average river flow data in these river basins. The SWAT+ model was also run for the period 2009 – 2016 using the EWEMBI forcing to compare with the WaPOR product, which has time series data in this period.

## 2.4 Scenarios setup

### 2.4.1 Climate scenarios

In this study, 31-year simulations were run for historical (1975 - 2005) and future (2070 - 2100) periods with one year warmup period. The meteorological forcing data was obtained through the Inter-Sectoral Impact Model Intercomparison Project (ISIMIP) phase 2b (Frieler et al., 2017). The forcing data was from 4 bias-adjusted global climate models (GCMs), namely

GFDL-ESM2M, HadGEM2-ES, IPSL-CM5A-LR, MIROC5 for both historical and future periods under Representative Concentration Pathways (RCPs) 2.6, 6.0 and 8.5. Thus, in total 24 simulations were conducted.

### 2.4.2 Land-use scenarios

Land-use scenarios were run using scenarios obtained from the Land-Use Harmonization Project phase 2 (LUH2; Hurtt et al., 2020). LUH2 reconstructs historical land uses from data based on the History of the Global Environment database (HYDE),



and multiple alternative scenarios of the future (2015–2100) from Integrated Assessment Models (IAMs; Hurtt et al., 2020). The scenarios used for land use were from three marker Shared Socioeconomic Pathway (SSP)-RCP combinations discussed in Hurtt et al. (2020). These scenarios include SSP1-RCP2.6, SSP4-RCP6.0 and SSP5-RCP8.5.

The SSP1-RCP2.6 scenario is derived from SSP1 baseline scenario where sustainable socioeconomic trends and ambitious climate policy result in reductions in agricultural land and increases in forest land (Doelman et al., 2018). Under SSP4-RCP6.0,

environmental policies are present in high- and medium-income countries only where afforestation is encouraged resulting in global increase cover by 3% from 2010 to 2100. There is also an increase in global crop and pasture land by 14% and 9% from 2010 to 2100, respectively. Under SSP5-RCP8.5, annual green house gas emissions more than double with very high levels of fossil use. Food demand is doubled and there is a strong expansion of global cropland with an increase of about 20% between 2010 and 2100 (Hurtt et al., 2020). More details about the scenarios SSP1-RCP2.6, SSP4-RCP6.0 and SSP5-RCP8.5

(henceforth referred to as land use scenarios for RCP 2.6, 6.0 and 8.5, respectively) are discussed by Hurtt et al. (2020).

We run another 24 runs for the CC combined with LULCC scenarios, making a total of 48 simulations when climate scenario simulations are considered. A list of scenario simulations (Including the run for ET evaluation) done in this study is shown in Table 1.

**Table 1: Simulations used in this study**

| Period | Scenario | Name | GCM Forcing | Land Use Scenario |
|---|---|---|---|---|
| 2009-2016 | - | ET-Eval | EWEMBI Reanalysis | - |
| | | | | |
| 1975 - 2005 | historical | CC-Historical | GFDL-ESM2M, HadGEM2-ES, IPSL-CM5A-LR, MIROC5 | - |
| 2070 - 2100 | RCP 2.6 | CC-RCP26 | GFDL-ESM2M, HadGEM2-ES, IPSL-CM5A-LR, MIROC5 | - |
| 2070 - 2100 | RCP 6.0 | CC-RCP60 | GFDL-ESM2M, HadGEM2-ES, IPSL-CM5A-LR, MIROC5 | - |
| 2070 - 2100 | RCP 8.5 | CC-RCP85 | GFDL-ESM2M, HadGEM2-ES, IPSL-CM5A-LR, MIROC5 | - |
| | | | | |
| 1975 - 2005 | historical | CC-LU-Historical | GFDL-ESM2M, HadGEM2-ES, IPSL-CM5A-LR, MIROC5 | historical |
| 2070 - 2100 | RCP 2.6 | CC-LU-RCP26 | GFDL-ESM2M, HadGEM2-ES, IPSL-CM5A-LR, MIROC5 | SSP1-RCP 2.6 |
| 2070 - 2100 | RCP 6.0 | CC-LU-RCP60 | GFDL-ESM2M, HadGEM2-ES, IPSL-CM5A-LR, MIROC5 | SSP4-RCP 6.0 |
| 2070 - 2100 | RCP 8.5 | CC-LU-RCP85 | GFDL-ESM2M, HadGEM2-ES, IPSL-CM5A-LR, MIROC5 | SSP5-RCP 8.5 |


The same vegetation cover as prescribed by LUH2 is kept. Thus, we are not considering dynamical vegetation shifts that might occur due to CC.

## 2.5   Model evaluation and analysis

The model was evaluated using monthly flow values from gauging station data obtained through the Global Runoff Data

Centre (GRDC, n.d.). Some gauging stations were not considered in our study as the data was on small rivers that were not represented at the model scale. Since the EWEMBI (Lange, 2016) forcing starts in 1979, data before 1979 was not usable and



most stations had data between 1980 and 1990 with extensive missing data after 1986. The same criteria as specified in Chawanda et al. (2020a; Section 2.3.2) was applied which resulted in 154 gauging stations. The WaPOR dataset (at 0.0022° × 0.0022° resolution) was used to check the model performance for ET both spatially and temporary.

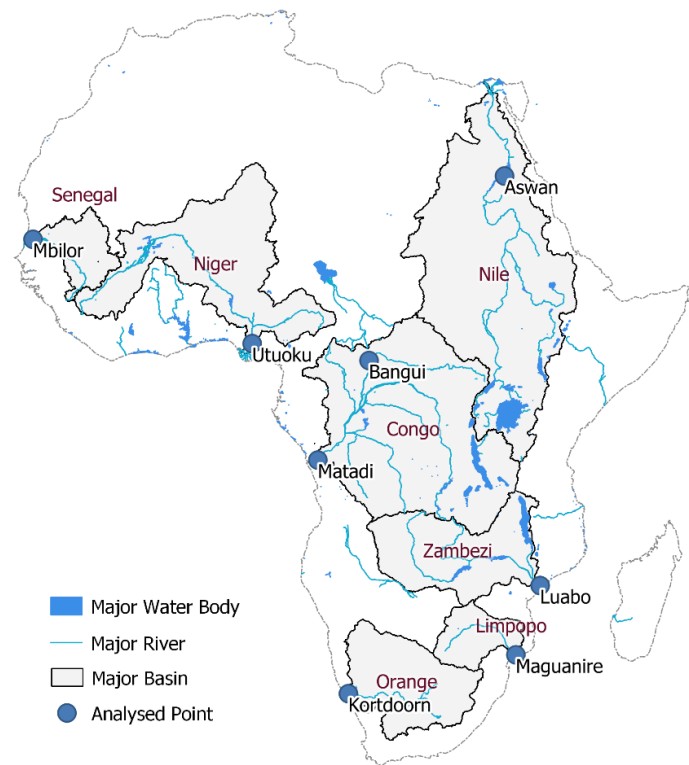


**Figure 1: Major river basins in Africa**

Analysis of projections for river flows was done for major river basins at the main outlets where possible (Figure 1) Characteristics of these river basins are shown in Table 2. We also evaluated the changes in river flows and ET in future climate in relation to historical climate.

**Table 2: Characteristics of the major river basins in Africa** (La et al., 2007; Lakshmi et al., 2018; Lange, 2016; Latrubesse et al., 2005; Lugomela et al., 2021; Nakayama, 2003)

| Basin | Nile | Senegal | Niger | Congo | Zambezi | Limpopo | Orange |
|---|---|---|---|---|---|---|---|
| Drainage Area ($10^6$ km$^2$) | 2.6 | 0.27 | 2.1 | 3.7 | 1.4 | 0.42 | 0.97 |
| Maximum elevation (m, rounded to 100m) | 4000 | 1200 | 2500 | 3200 | 2900 | 2200 | 3300 |
| Annual Accumulated Precipitation (mm) | 337 | 500 | 650 | 1,600 | 956 | 550 | 360 |
| Daily Average Temperature (℃) | 27 | 24 | 24 | 25 | 22 | - | 25 |
| Mean Discharge (m$^3$/s) | 1,584 | 680 | - | 40,900 | 3,511 | 170 | 365 |





## 3 Results

### 3.1 Model performance in the historical Period

#### 3.1.1 Simulation of river flows

The performance of the uncalibrated model with irrigation and reservoirs as measured by NSE values was generally poor. HMBC improved NSE values in 98 gauging stations whereas 38 stations did not have changes in performance after HMBC. 18 stations had a lower performance after HMBC than before HMBC (Figure 2).

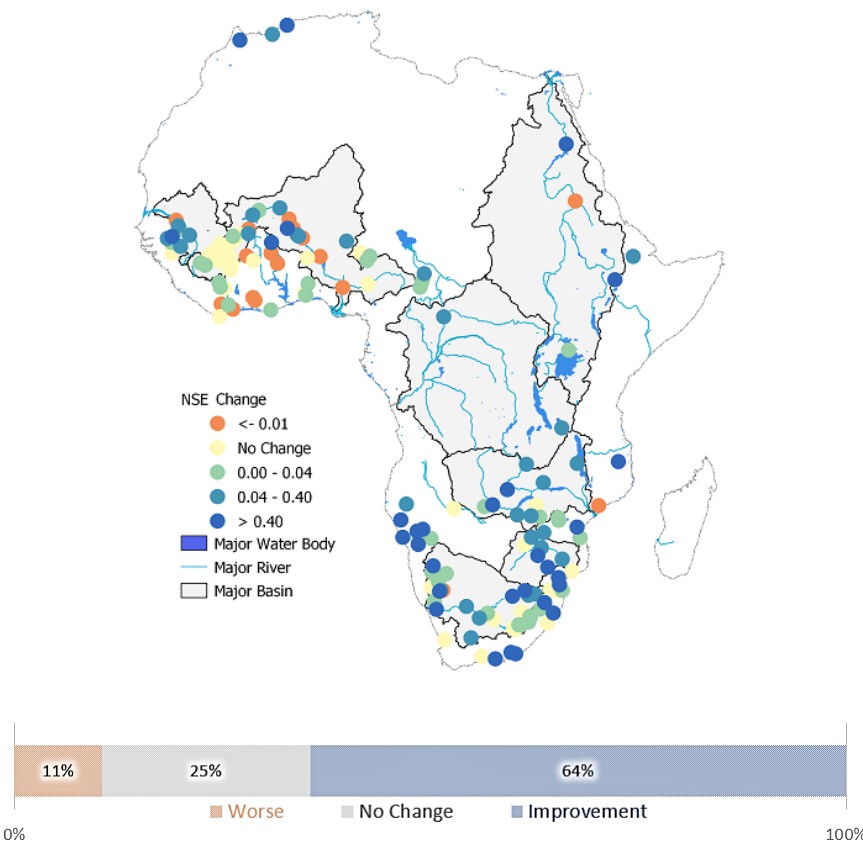


**Figure 2: (a) Changes in NSE values across the gauging stations in the SWAT+ Africa model after performing Hydrologic Mass Balance Calibration (HMBC). (b) Distribution of changes in NSE values after HMBC.**

After performing HMBC, 96 out of 154 gauging stations had a monthly NSE value > 0 with 50 Gauging stations having monthly NSE above 0.5 (Figure 3).





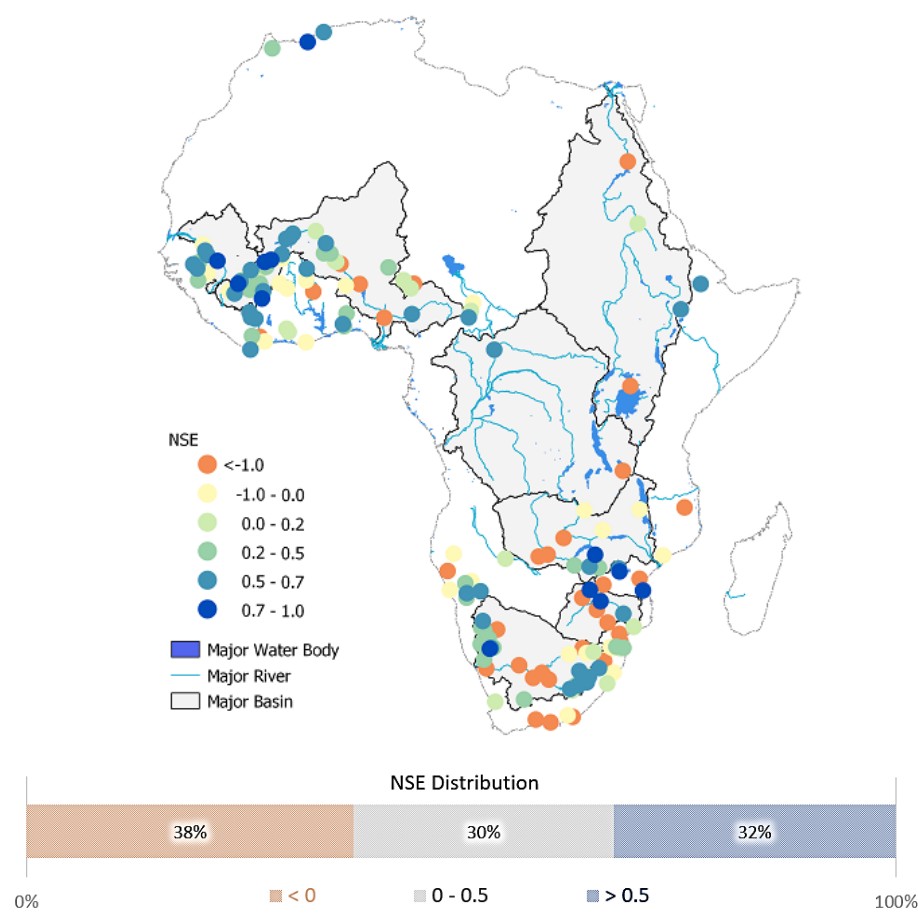


**Figure 3: (a) Monthly NSE values for gauging stations across the SWAT+ Africa Model. (b) Distribution of NSE values after HMBC (Evaluated for river flows between 1980 - 1986).**

Low model performance was observed in gauging stations that were downstream of reservoirs (Figure 4). This was expected,

as a lack of data on dam management contributes to a poor simulation of river flows through reservoirs (Chawanda et al., 2020a). The poor performance downstream of reservoirs also had an impact on the performance on the most-downstream gauging stations of major river basins as observed in the Orange, Niger, Nile (NSE < 0) and Limpopo (NSE 0.04) whereas in the Senegal River basin, where there are no reservoirs implemented along the main river, an NSE of 0.55 was achieved.

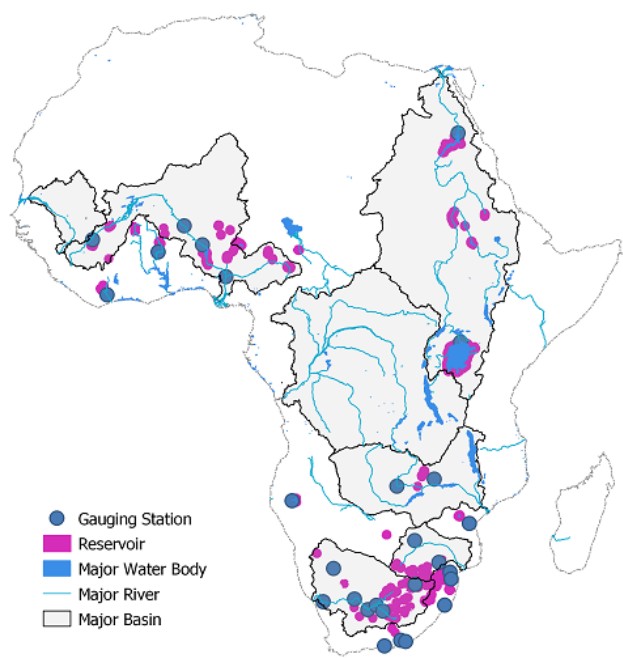

**Figure 4: Gauging stations that had NSE < 0 with a reservoir upstream. (The size of the reservoirs has been exaggerated to make the location of the reservoirs easily visible).**

### 3.1.2 Simulation of evapotranspiration

Model ET was comparable to WaPOR ET spatially using the *ET-Eval* run (Table 2.1). The model captured high ET in coastal zone of North Africa, West and Central Africa and the Ethiopian Highlands.

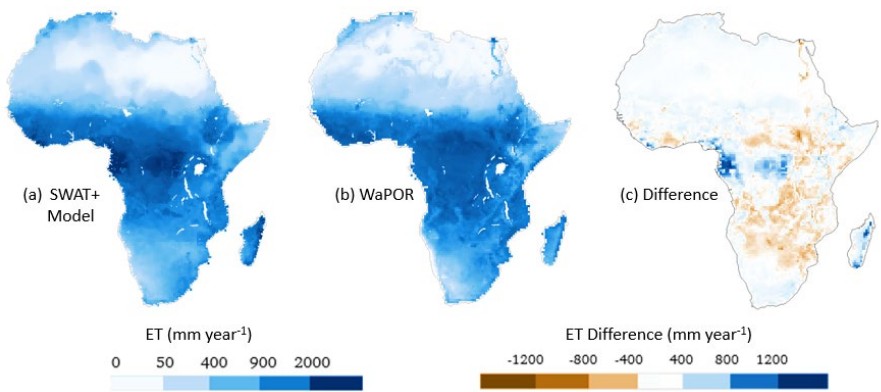


**Figure 5: Annual mean ET from (a) the SWAT+ model and (b) WaPOR reference product. (c) difference between the SWAT+ model and WaPOR (SWAT+ model– WaPOR) for the period 2009 – 2016.**

The model also captured the low ET values expected in the Sahara, Namib and Kalahari Deserts (Figure 5 a and b). However, the SWAT+ model overestimated ET in the Congo Basin by 58 mm year$^{-1}$ and underestimated annual ET in the lower Nile




River by 42 mm year⁻¹. The high observed ET values (locally up to 500 mm year⁻¹) in the lower Nile, which the model underestimates, are expected due to irrigation activity and multiple cropping sessions in the area. The model underestimated the ET in the Ethiopian Highlands by 50 mm year⁻¹, and ET was also underestimated in parts of the greater Horn of Africa. The SWAT+ model overestimates ET in Madagascar by 360 mm year⁻¹ (Figure 5 c), but the highest overestimation was in the West of the Congo River basin caused by excessive rainfall in the area (Figure 5). In temporal terms, the model captures the

magnitude of the overall ET values in Africa but slightly underestimates the annual average ET by 10 mm year⁻¹ (1.8 %) on average.

## 3.2    Model projections

### 3.2.1    Projected changes in precipitation

The spatial patterns of projected changes in precipitation as simulated by the driving ISIMIP2b GCMs show an overall increase

in the west of the Congo River basin under all RCPs. The White Nile River Basin, the Ethiopian Highlands, and the Horn of Africa receive higher precipitation in the future under all RCPs. On average, the White Nile River Basin receives 1046 mm for RCP 2.6  1136 mm for RCP 6.0, and 1253 mm for RCP 8.5  in contrast to 1014 mm in the historical period. The Ethiopian highlands have the highest precipitation increase (392 mm spatial average) under RCP 8.5 while upper Madagascar has the largest decrease of 226 mm under RCP 8.5. In general, the changes in precipitation are more pronounced under RCP 6.0 than

RCP 2.6 and under RCP 8.5 more than under RCP 6.0 (Figure 6).

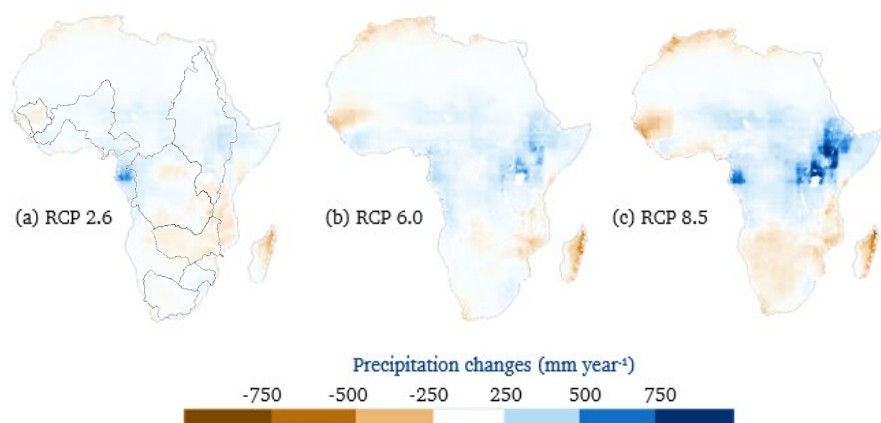

**Figure 6: Annual mean precipitation changes for the future period (2071 - 2100) under (a) RCP 2.6, (b) RCP 6.0 and (c) RCP 8.5 averaged across the four ISIMIP2b GCMs. Maps are calculated by subtracting the average precipitation for the future period from the average precipitation for the historical period (1976 - 2005).**

Southern Africa is projected to experience an overall decrease in future precipitation, including Madagascar and the severity of the decrease scales with the emission scenario. The decrease in future precipitation for Madagascar is more severe than Southern Africa under all scenarios.



### 3.2.2     Projected changes in evapotranspiration

Under RCP 2.6, CC alone leads to an increase in the ET values between 7°S and 15°N (Figure 7) according to the SWAT+
projections with an overall decrease in ET in West Africa and Southern Africa (including Madagascar) under all RCPs. The
increases become more pronounced moving from RCP 2.6 to RCP 8.5 (Figure 8).

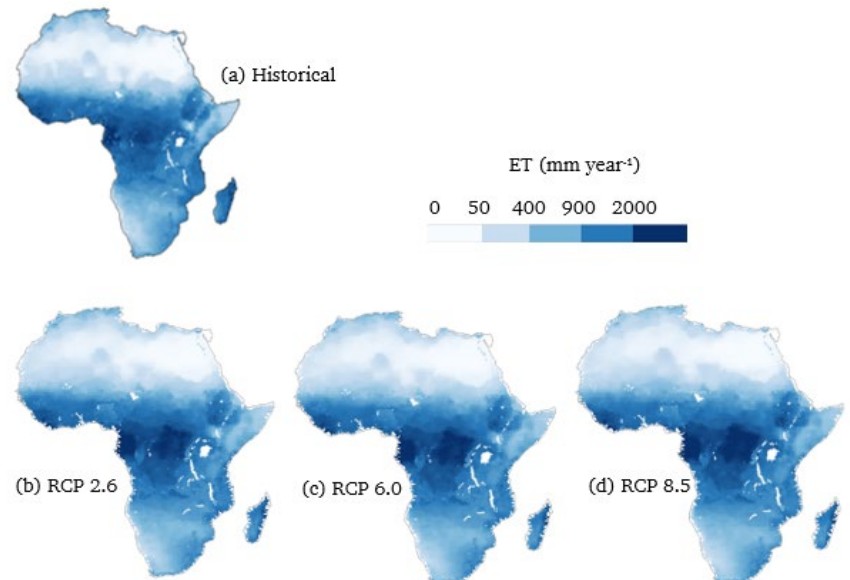

**Figure 7: Annual mean ET for (a) Historical period (1976 - 2005) and future period (2071 - 2100) under (b) RCP 2.6, (c) RCP 6.0
and (d) RCP 8.5 for CC.**

The increases in precipitation in the Ethiopian highlands are matched by an increase in ET in the same area (Figure 8) under
all RCPs under CC. The central parts of Africa also show a similar signal for both precipitation and ET under CC. There is
also a strong ET decrease in the Senegal River basin and coastal North Africa compared to the region south of the Niger River
basin (4.5 - 13.5° N, -7.0 – 2.0° E) and most of the region south of the 7°S latitude. In contrast, a large increase in ET is
projected over the Ethiopian Highlands and parts of the Upper White Nile River basin averaging up to 46 mm year[-1] under
RCP 2.6 and 160 mm year[-1] under RCP 8.5 (Figure 8 b and c).

In contrast to the pure CC effect, the ET under combined effects of CC and LULCC in the Congo River basin decreases in the
future, especially for RCP 2.6 (-207 mm year[-1] change for CC & LULCC vs 96 mm year[-1] change for CC) Figure 8 d, e, and
f). The role of the CC is also apparent under CC & LULCC. Under CC & LULCC, there is a larger decrease in ET under RCP
2.6 (Figure 8 d) than under RCP 8.5 (Figure 8 f), while there is a smaller increase in ET under pure CC in RCP 2.6 (Figure 8
a) compared to RCP 8.5 (Figure 8 c).

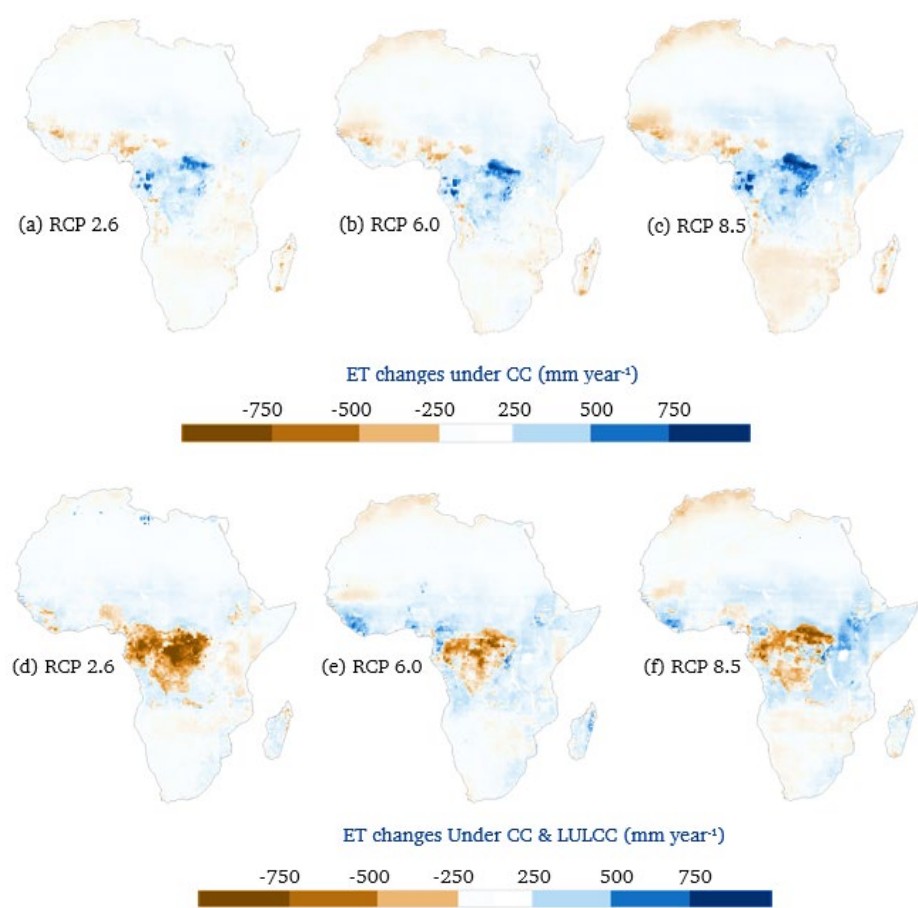

**Figure 8: Annual mean ET change for the future period (2071 - 2100) under (a) RCP 2.6, (b) RCP 6.0 and (c) RCP 8.5 for CC and (d) RCP 2.6, (e) RCP 6.0 and (f) RCP 8.5 for CC & LULCC (Maps are calculated by subtracting ET for the future period by ET for the historical period (1976 - 2005)).**

Madagascar shifts from lower ET under CC alone (-42 mm year$^{-1}$, -65 mm year$^{-1}$ and -59 mm year$^{-1}$ for RCPs 2.6, 6.0 and 8.5, respectively) to increased ET under CC & LULCC (44 mm year$^{-1}$, 92 mm year$^{-1}$ and 26 mm year$^{-1}$ for RCPs 2.6, 6.0 and 8.5, respectively). There is also a shift within the CC & LULCC scenarios from a negative change in ET under RCP 2.6 to a positive change in ET under RCPs 6.0 and 8.5 for parts of East Africa, Liberia and Guinea (10.2 -11.1° N, 13 - 9° W).

### 3.2.3 Projected changes in water availability

The inclusion of the land-use scenarios modifies the signal of change in water availability (here defined as precipitation – ET) in the Congo River basin. There is an overall reduction in water availability under CC of -79 mm year$^{-1}$, -64 mm year$^{-1}$ and -57 mm year$^{-1}$ under RCPs 2.6, 6.0, and 8.5, respectively while there is an increase of over 120 mm year$^{-1}$ in available water under CC and LULCC (Figure 9).

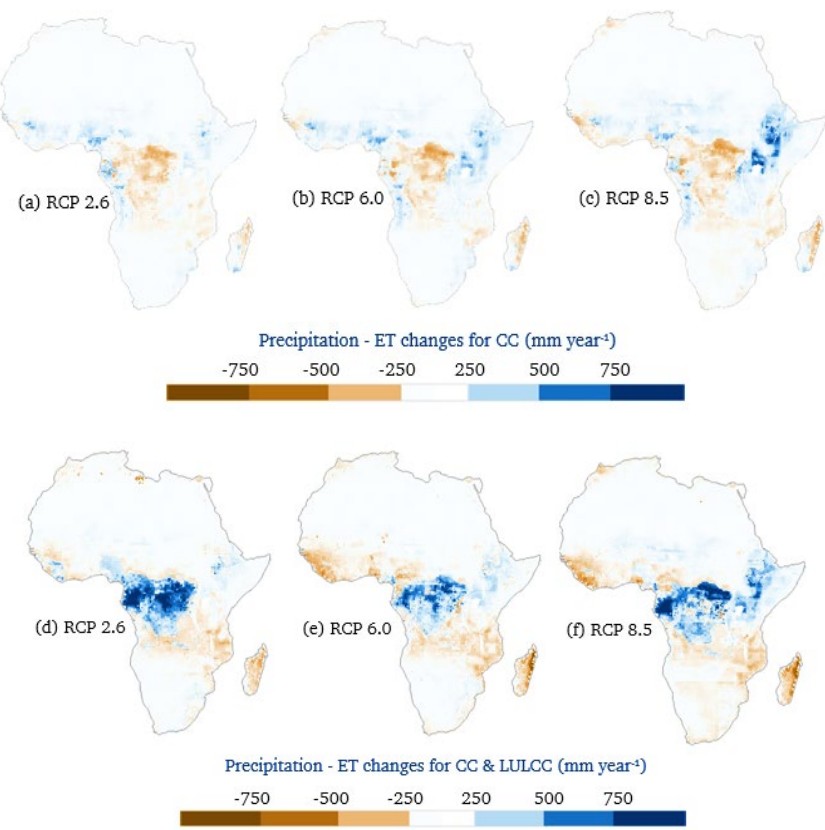

**Figure 9: Annual mean water availability (P-ET) change for future period (2071 - 2100) under (a) RCP 2.6, (b) RCP 6.0, and (c) RCP 8.5 for CC and (d) RCP 2.6, (e) RCP 6.0 and (f) RCP 8.5 for CC & LULCC. (Maps are calculated by subtracting average precipitation - ET for the future period by average precipitation - ET for the historical period (1976 - 2005)).**

In West Africa, an increase in water availability is projected under all RCPs under CC alone. On average, water availability in the Niger River basin increases by 44 mm year[1] under RCP 2.6, 50 mm year[1] under RCP 6.0, and 37 mm year[1] under RCP 8.5. There is also a 22 mm year[1] increase in water availability in the Niger River basin under RCP 2.6 if both CC and LULCC are considered but a reduction of -25 mm year[1] and -3 mm year[1] for RCP 6.0 and RCP 8.5, respectively.

In Senegal, the increase in water availability under RCP 2.6 for CC only scenario is caused by a small decrease in precipitation in future climate that is accompanied by an even smaller decrease in ET as a response, leading to an excess in the change in water availability. RCPs 6.0 and 8.5 also show a decrease in both precipitation and ET, but there is a deficit in water availability by 8 mm year[-1] for RCP 6.0 and 11 mm year[-1] for RCP 8.5 under CC. When both CC and LULCC are considered in the Senegal Basin, all RCPs show more decrease in water availability. Thus, in West Africa, combining CC and LULCC scenarios reduces the water availability values relative to the CC only scenario.

The Orange River basin is projected to experience an increase in water availability under CC under RCPs 2.6 and 6.0 while for RCP 8.5 where there is a reduction of -13 mm year[1]. In contrast, Zambezi has a negative signal under all RCPs in CC



averaging -22 mm year[1] and under CC & LULCC (except RCP 8.5) averaging -39 mm year[1]. With a decrease in precipitation in Madagascar under all CC RCPs, there is a reduction in the water availability under CC, but there is even more reduction

when LULCC is considered (Figure 9).

### 3.2.4    Projected changes in river flows

River flows under CC were reduced by the inclusion of LULCC in Senegal, Niger and Orange River basins while river discharge increased in the Congo River basin (Figure 11). Limpopo, Zambezi and Nile had either an increase or a decrease of river flows depending on RCP. Overall changes in average river flows from the major river basins were within -30% to 30%

for all RCPs except for the Nile and the Niger River basins, where annual average river flows are projected to increase by over 70% under RCP 6.0 and by over 100% under RCP 8.5.

A closer look revealed that simulation CC-RCP60 (Table 1) with the IPSL-CM5A-LR GCM had a reservoir bug where unrealistic discharge 1.38E+11 $m^3$ $s^{-1}$ was released on in May, 2086 from the Nile reservoir at Owen Falls. Another reservoir failed in the Niger River basin for simulation CC-RCP85 with MIROC5 GCM releasing 1.38E+13 $m^3$ $s^{-1}$ in August, 2084.

With these cases of reservoir simulation instabilities, IPSL-CM5A-LR GCM was excluded for the Nile River basin, and MIROC5 GCM was not included for the Niger River basin. Figure 12 characterises projections under pure CC and projections under combined CC and LULCC .

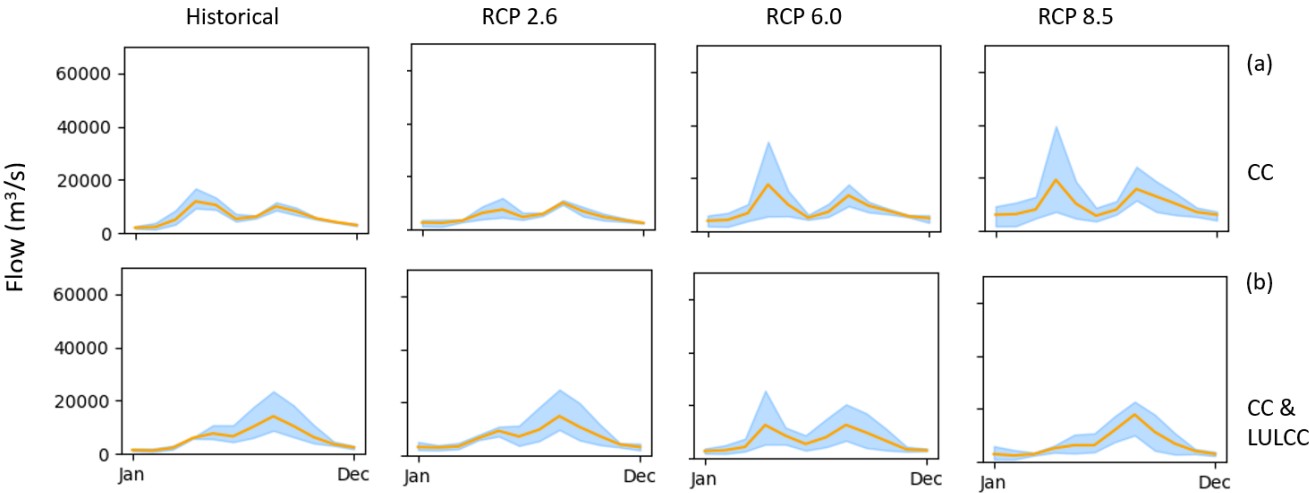

**Figure 10: Mean monthly river flows averaged across GCMs for the Nile Basin outlet at Aswan (lat: 22.7° N, lon: 32.50° E) for**
**historical (1975 - 2005), RCP 2.6, RCP 6.0 and RCP 8.5 (2070 - 2100) under (a) CC and (b) CC & LULCC. (Orange line represents mean while the sky-blue band represents the range of values across GCMs).**

In the Nile, peak river flows are simulated in March, April, May and August, September and October. Under RCP 2.6, there is a general decrease in monthly river flows under CC in the Nile driven by a 15% reduction of river flows in the peak flow months. In contrast, there are 28% and 50% increases in the same months under RCP 6.0 and RCP 8.5 (Figure 11 a). The





monthly river flow changes under CC & LULCC were less than under pure climate except under RCP 2.6, where there was an

increase in projected river flows (Figure 11 b).

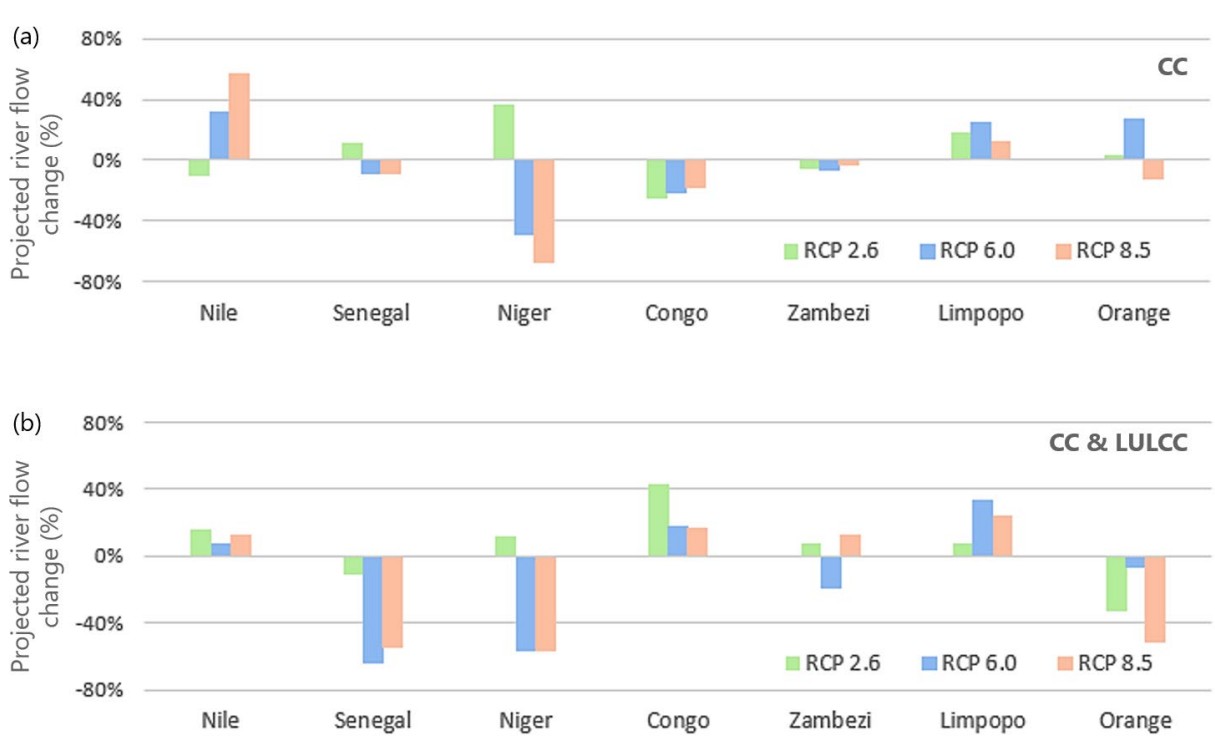

**Figure 11: Projected change in river flow (%) for the future period (2071 - 2100) relative to the historical period (1976 - 2005)**
**obtained from multi-model mean of average annual river flows at the outlet of major basins in (Figure 1) (The percent change for**
**the Nile was calculated just before Lake Nasser).**





**Figure 12: 10th, 5 0th and 90th percentile river flows across major river basins in Africa under CC and under CC & LULCC for the historical (1975 - 2005) and RCP 2.6, RCP 6.0 and RCP 8.5 (2070 - 2100).**

In the Senegal basin, there was only a slight change in the mean monthly river flows under CC (Figure 11 a). However, the spread of values across GCMs increased from historical to RCPs 2.6, 6.5 and 8.5. Also, the river flows in September are, on average, greater than those in August under RCP 8.5, which is different when compared to the historical period and the other RCPs (Figure 13). When LULCC is considered, the mean monthly river flows in the wet months of July to October decrease towards the future (Figure 13), leading to an overall decrease in river flows under CC & LULCC (Figure 11).

In the Niger Basin, the upper part of the basin has different dynamics from the lower part. Looking at the outlet at Amgoundji (17.06° N; 1.12° W), there is an increase in river flows during the wet months (August to December) under RCPs 2.6, 6.0 and 8.5 by 61%, 71% and 58% of the historical period river flows, respectively, for the same months under CC. Accounting for LULCC in addition to CC leads to a reduction of average discharge under RCPs 2.6 and 6.0 by 48% and 4% but and increase under RCP 8.5 by 7%.


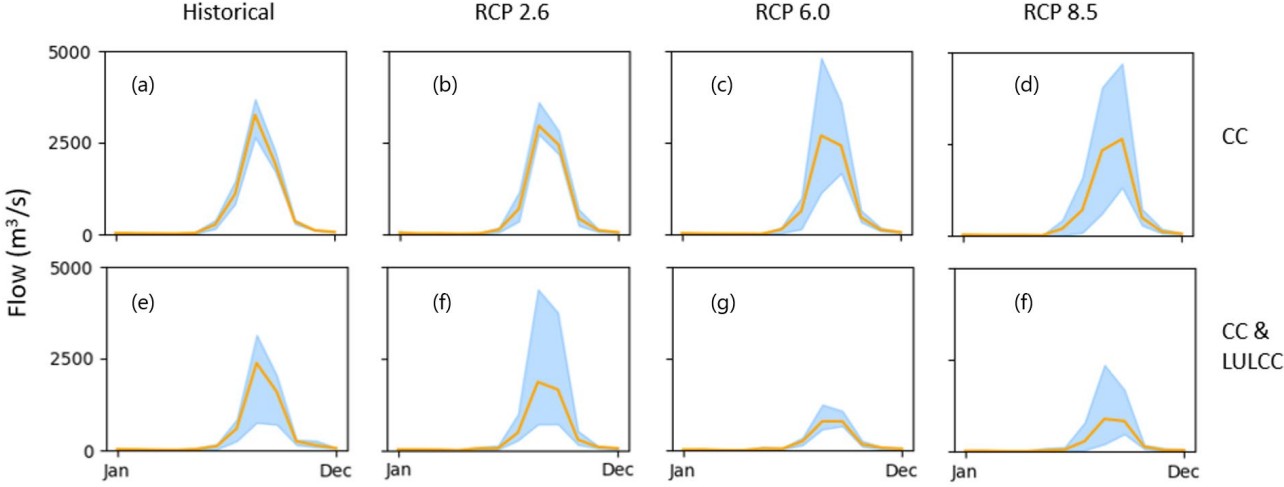

**Figure 13: Mean monthly river flows averaged across GCMs for the Senegal Basin outlet at Mbilor (16.58° N, 15.61° W) for historical (1975 - 2005), RCP 2.6, RCP 6.0 and RCP 8.5 (2070 - 2100) under (a) CC and (b) CC & LULCC. (Orange line represents mean while the sky-blue band represents the range of values across GCMs).**

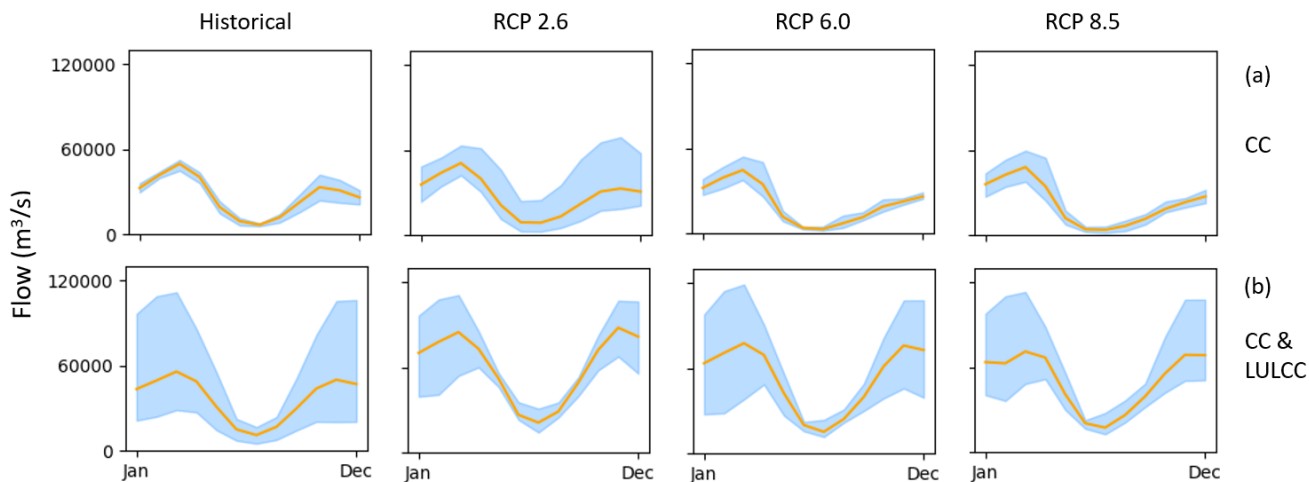


**Figure 14: Mean monthly river flows averaged across GCMs for the Congo Basin outlet at Matadi (lat: 5.80° S, lon: 13.40° E) for historical (1975 - 2005), RCP 2.6, RCP 6.0 and RCP 8.5 (2070 - 2100) under (a) CC and (b) CC & LULCC. (Orange line represents mean while the sky-blue band represents the range of values across GCMs).**

The mean monthly river flows in the Congo River basin show an overall decrease under all CC scenarios compared to the

historical period (Figure 11 a). The decreases are spread throughout both wet and dry months of the year. There is also an increase in the spread of values across GCMs at the outlet of the Congo River basin. Under CC & LULCC, there is more inter-annual variability in river flows within the GCMs for all scenarios (Figure 14).





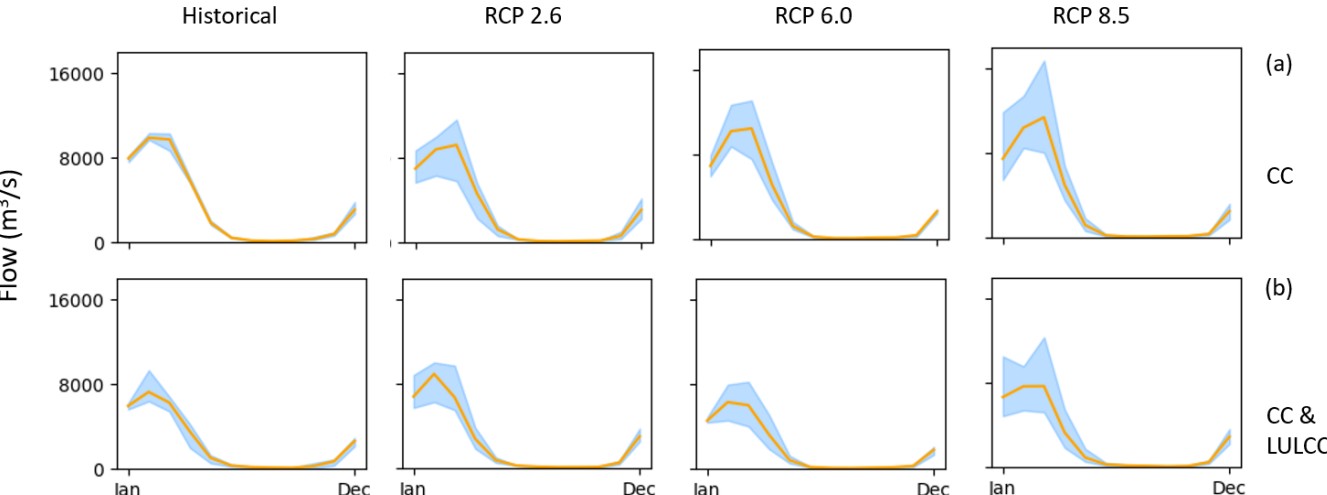

**Figure 15: Mean monthly river flows averaged across GCMs for the Zambezi Basin outlet at Luabo (lat: 18.40° S, lon: 36.10° E) for historical (1975 - 2005), RCP 2.6, RCP 6.0 and RCP 8.5 (2070 - 2100) under (a) CC and (b) CC & LULCC. (Orange line represents the mean while the sky-blue band represents the range of values across GCMs).**

The average river flows for the Zambezi River in the historical period are similar in both CC and CC & LULCC, but there is an increase in river flows projected under CC & LULCC in future scenarios. The changes in mean monthly river flow at the outlet of the Zambezi River basin, Luabo (lat: 18.40° S; lon: 36.08° E), were within 10% on average (Figure 11), but the range of the spread of values across GCMs increases from historical, RCP 2.6, RCP 6.0 to RCP 8.5 (Figure 15). With LULCC, the river flows in Zambezi show an increase under RCPs 2.6 and 8.5 (Figure 11), but this is because of lower simulated historical values under LULCC (Figure 15 b).

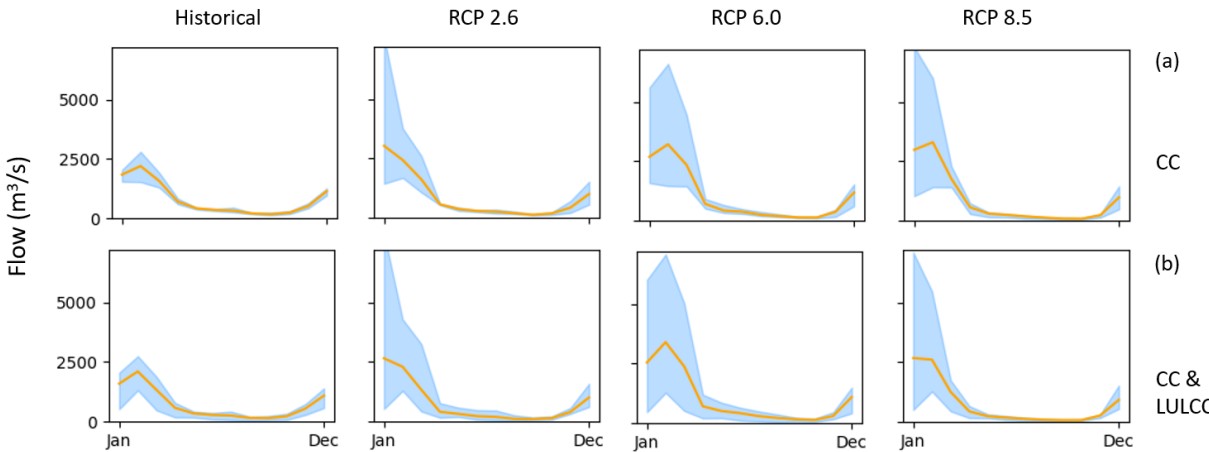

**Figure 16: Mean monthly river flows averaged across GCMs for the Limpopo Basin outlet for historical (1975 - 2005), RCP 2.6, RCP 6.0 and RCP 8.5 (2070 - 2100) under (a) CC and (b) CC & LULCC. (Orange line represents the mean while the sky-blue band represents the range of values across GCMs).**





The Limpopo River basin is projected to experience a slight increase in both the monthly mean river flows in the wet months (January to April), and a larger variability of the river flows from the historical periods to future periods under CC. Under CC and LULCC, river flows increases from CC scenarios under all RCPs but RCP 6.0 has the largest increase in monthly flow during the wet months of up to 33% (Figure 16).

For the Orange River basin, an increase in average mean monthly river flows is projected in the wet months (October to April) across GCMs for CC under RCP 2.6 and RCP 6.0 by 4% and 32%, respectively, while river flow in RCP 8.5 demonstrates a decrease in the wet months (Figure 17). The spread in values across GCMs also increases relative to the historical period, with the largest change observed under RCP 6.0. Under both CC and LULCC, the monthly river flows in the river basin decrease from the CC river flows across all RCPs (Figure 11).

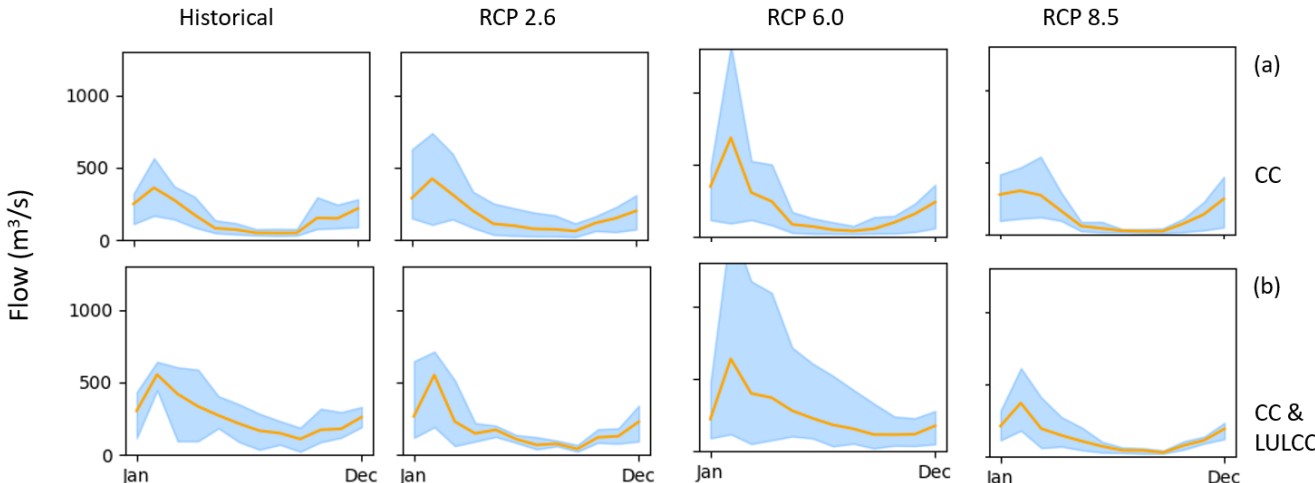

**Figure 17: Mean monthly river flows averaged across GCMs for the Orange Basin outlet for historical (1975 - 2005), RCP 2.6, RCP 6.0 and RCP 8.5 (2070 - 2100) under (a) CC and (b) CC & LULCC. (Orange line represents the mean while the sky-blue band represents the range of values across GCMs).**

## 4  Discussion

### 4.1  Model performance of the SWAT+ Africa model after HMBC.

The model was calibrated using HMBC following Chawanda *et al*. (2020). There was an improvement in the performance of river flows following calibration (Figure 2), yet the performance of the model in gauges downstream of reservoirs is consistently poor (Figure 4). This is mainly due to the difficulty in modelling human control of river discharge at dams holding the water in most reservoirs on the continent. The implementation of generalised decision tables for various reservoir types in this study does improve model performance, but more data is needed for a more accurate representation of reservoir control.

HMBC's ability to target specific components of the hydrological cycle improves the model's ability to capture the average annual values of ET in Africa when compared to the WaPOR dataset, although it slightly underestimated ET in years 2015





and 2016. The model was able to capture the spatial pattern of ET (Figure 5). The ability of the model to have a comparable

spatial pattern in simulated ET relative to WaPOR ET is in part due to the implementation of irrigation. High ET is observed

in areas where irrigation was implemented. In some regions, such as the lower Nile, the model underestimated ET, which may

point to under-irrigation, but the values for irrigation simulated by SWAT+ in the Nile Delta are in line with data collected by

Kubota *et al*. (2020), ranging from 320 mm to 450 mm. Thus, the underestimation may be attributed to having a single growing

season implemented in the model, which hinders further transpiration from the irrigated fields (Nkwasa et al., 2022). Irrigation

fields are also very fragmented on the continent due to small-scale agriculture along riverbanks which is not usually reflected

in global datasets and poses a challenge in the representation of irrigation in models at such a large scale. As such, there is

room for improvement in irrigation datasets and representation of irrigation in large-scale modelling.

In this study, application of HMBC was limited in the Nile and Congo River basins due to lack of gauging stations with river

flow data. Thus, HMBC was applied only for ET in some calibration zones. Furthermore, as with most ET products, the

reference ET product used (WaPOR) is a result of a 'model' which limits the improvements to model performance in some

areas.

## 4.2    Impacts of climate change.

The increase in precipitation over the Ethiopian Highlands (Figure 6) is reflected in the river flows for the Nile River basin

(Figure 11). The increase in precipitation was slightly offset by an increase in ET over the Ethiopian Highlands area (Figure

8) which takes away from water available for surface runoff and infiltration. However, there is still a significant increase in

the available water for the Upper Nile in the future period (Figure 9), leading to overall increases in river flows along the entire

Nile River. In their study to find impacts of CC in the Nile Basin, Di Baldassarre et al. (2011) show an ensemble of models

that project changes in river flows at Diem in Upper Blue Nile ranging from -67% to 55% by 2098. The disagreements in the

signal of future annual river flows point to high uncertainty in future river flows for the river basin. However, Conway (2017)

argues that the river flows in the Nile are more likely to increase in the future.

The results also indicate an increase in variability of flows in the Nile River under CC (Figure 10). This is in line with findings

from other studies, such as that by Siam and Eltahir (2017), which project up to 50% increase in interannual variability. The

Nile River is important geo-politically since countries upstream and downstream of the river have to work together in managing

optimal river flows for all interested parties. While the construction of the Grand Ethiopian Renaissance Dam may reduce the

river flows downstream permanently (Ramadan & Negm, 2013), such a dam could act as a buffer for the increased extreme

events (Sterl et al., 2021). In addition, an overall increase in river flows due to CC would offset flow decreases after the

construction of the Dam.

The results show that precipitation is likely to decrease in the Senegal River basin due to CC. This has the potential of extending

the Sahara further down. Projections show an overall decrease in river flows in the future under CC. The future drying up of

Senegal, in addition to sea-level rise (Croitoru et al., 2019), has important implications in the area. Roughly 70% of the





population in the river basin earns a living through agriculture (Kohli & Alam, n.d.). Saltwater intrusion into aquifers and arable land, which has been observed in the area (Delphine, 2013), is likely to intensify under CC, which would further devastate the agricultural sector and hence the livelihood of the population of the Senegal River basin.

Projections for the Niger river basin show an increase in overall river flows under RCP 2.6 and a decrease in river flows and available water if the outlier GCM for the area (MIROC5) is not accounted for. The same signals were projected in the Senegal River basin. West Africa has experienced a southward shift in climatic zones (Wittig et al., 2007), leading to drier conditions in the northern parts of the region (IPCC, 2021). The projected drying of the Niger River basin together with the Senegal River basin implies further desertification of the northern parts of West Africa, which will further expand the Sahara Desert.

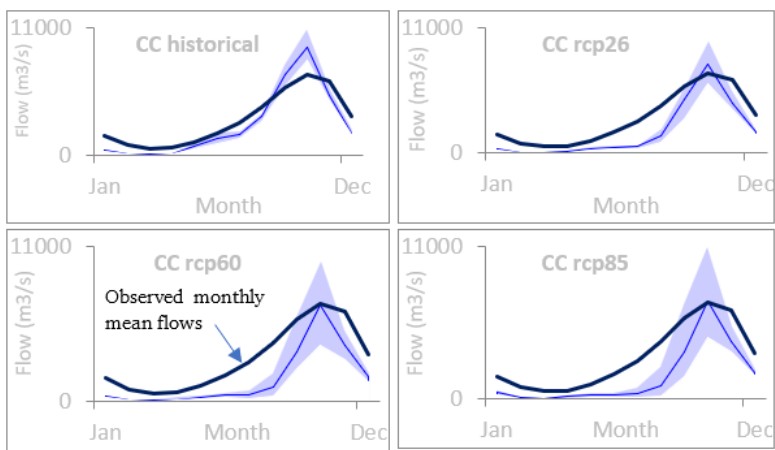

**Figure 18: Mean monthly river flows averaged across 4 GCMs for Ubangi River at Bangui (Congo River basin) under historical period (1976 - 2005), and in the future period (2071 - 2100) under RCP 2.6, RCP 6.0 and RCP 8.5.**

The projections show a decrease in the river flows from the Congo River basin and Zambezi River basin (Figure 11). The decreases in the Congo River basin flows can be attributed to a decrease in river flows during wet months. This is apparent at Bangui (18.58 E, 4.36 N) (Figure 2.1) along the Ubangi River (Figure 18) but is not apparent for the main outlet of the Congo River basin (Figure 14). The overall decrease in river flows in the Zambezi River basin is attributed to a decrease in river flows during dry months even though there was an increase in peak river flows in wet months for all RCPs. This is in line with the relative changes in precipitation and ET in the basin. There is a larger drop in future rainfall compared to the drop in future ET which leads to an increased deficit in available water (Figure 19 b). Unlike in the Zambezi River basin, the overall decrease in water availability in the Congo is due to a higher increase in ET than the increase in precipitation for the future period (Figure 19 a).





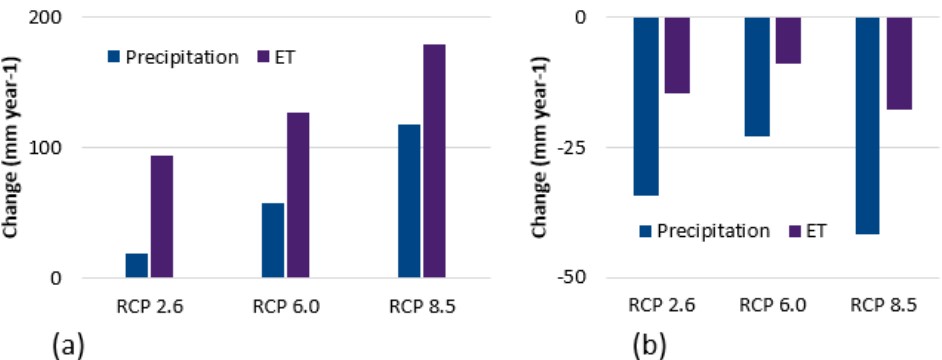

**Figure 19: Changes in spatially averaged annual ET vs Changes in spatially averaged annual Precipitation in the (a) Congo River basin and (b) Zambezi River basin, under CC for simulated scenarios for the future period (2071 - 2100) relative to the historical period (1976 - 2005).**

Unlike in the Congo River basin, the river flows in Zambezi River basin do not change much but show more variability in the future CC scenarios than in the historical period. The increased variability in river flows was also observed in the Limpopo River basin with an increase in overall river flows from the catchment. This is because the change in water availability increases in the future for RCPs 2.6, 6.0 and 8.5 by 14 mm year[1], 18 mm year[1], and 7 mm year[1], respectively. The increase is, however, observed in wet months (December to April), but there is still a slight decrease of flow in the dry months.

The increased flow variability in both Zambezi and Limpopo could mean increased occurrence of extreme events such as more floods and droughts. With agriculture being the main source of livelihood in the basins, crop failure due to drought or flooding could leave millions in need of food aid. This calls for coordination between the countries that share the basins to combat impacts of future climate variability.

The behaviour observed in the Senegal River basin under RCP 2.6 is also observed in the Orange River basin for RCPs 2.6 and 6.0, while the behaviour observed in the Senegal River basin under RCPs 6.0 and 8.5 is also observed in the Orange River basin under RCP 8.5. Herring *et al*. (2018) pointed out that the prevalence of droughts in South Africa has tripled in the last 60 years due to CC. The occurrence of droughts is likely to increase with the projected future decrease in precipitation in the Orange River basin (Figure 6). This is in line with a study by Pokhrel *et al*. (2021), who predict a general increase in drought hazards in the southern hemisphere. Thus, agricultural production is likely to suffer due to insufficient rains as more droughts are to be expected in the future. The model suggests a slight increase in available water under CC in the Orange River basin, but this is due to ET being very sensitive to the decrease in precipitation.

All projections are characterised by a wider spread in flow values across GCMs in the future, which is caused mainly by enhanced climate variability in each of the GCMs.



## 4.3     Impacts of land-use change

The combined impacts of CC and LULCC on ET are more pronounced under all RCPs compared to the pure CC impacts (Figure 8). LULCC had the strongest impact in the Nile, Congo, Senegal and Orange River basins. The Congo River basin shows a strong decline in ET under RCP 2.6 with mild changes in ET all over the continent under combined drivers. RCPs 6.0 and 8.5 show a weaker decrease for ET in the Congo basin compared to RCP 2.6 but a stronger signal elsewhere. The reduction in ET in the Congo Basin is mainly driven by projected deforestation under all RCPs (see Figure 20 a for RCP 8.5). Both the reduction of ground cover and reduced ET lead to increased surface runoff (Figure 11 b).

With deforestation in mind, precipitation is expected to reduce in the future. Dyer *et al.* (2017) concluded that a significant proportion of the moisture responsible for rainfall in the Congo River basin comes from the Indian Ocean (which may also be amplified under CC) and thus, deforestation is not expected to drive precipitation changes in the area. For the moisture that is recycled within the basin, it is likely that the increased precipitation due to greenhouse gas forcing is more important than the decrease in ET due to deforestation.

In West Africa, especially Senegal, increased agricultural activity is responsible for the future increase in ET under LULCC in the area but with projected precipitation, Available water is expected to decrease further than that under CC alone. This makes the situation for agriculture in the Senegal River basin at a higher risk if LULCC is considered. This is in addition to saltwater intrusion and a potential increase in droughts. A similar situation is observed in Madagascar where increased agricultural activity reduces the water availability in the future by promoting evapotranspiration under LULCC. With projected decrease in precipitation, water stress is expected to become an issue. However, no literature documenting same saltwater intrusion problems and loss of costal lands due to sea level rise experienced in Senegal was found.

LULCC has minimal impact in the Zambezi and Limpopo River basins due to mild changes in future scenarios of LULCC. If these LULCC scenarios are realised, climate variability could be a more important issue in the Zambezi and Limpopo basins unlike in Congo or Senegal Basins.

In the Orange River basin, there is an increase in agricultural land in the east side of the catchment where a significant proportion of the water comes from. This change in LULCC in addition to future precipitation decreases reduces the water availability in the area under CC & LULCC and hence a reduction of river flows at the outlet of the river basin.

Studies focusing on the impacts of LULCC on hydrological indicators such as ET and surface runoff in Africa often look at historical impacts and are done at very small scale e.g. Warburton *et al.* (2012) and Yira *et al.* (2016). Very few studies investigated hydrological effects of future land-use change. For example, Näschen *et al.* (2019) looks at land impacts of LULCC until 2030 on water resources but the differences in LULCC development and the small scale nature of the study does not allow direct comparison of results. However, such studies highlight that LULCC can have substantial effects on hydrological indicators.



The results of CC & LULCC projections also show artefacts in the ET and water availability change maps. These are caused by artefacts in either the land-use scenarios that were used (Figure 20 a) or the weather forcing (Figure 20 b). These artefacts were also observed in Chawanda et al. (2020a).

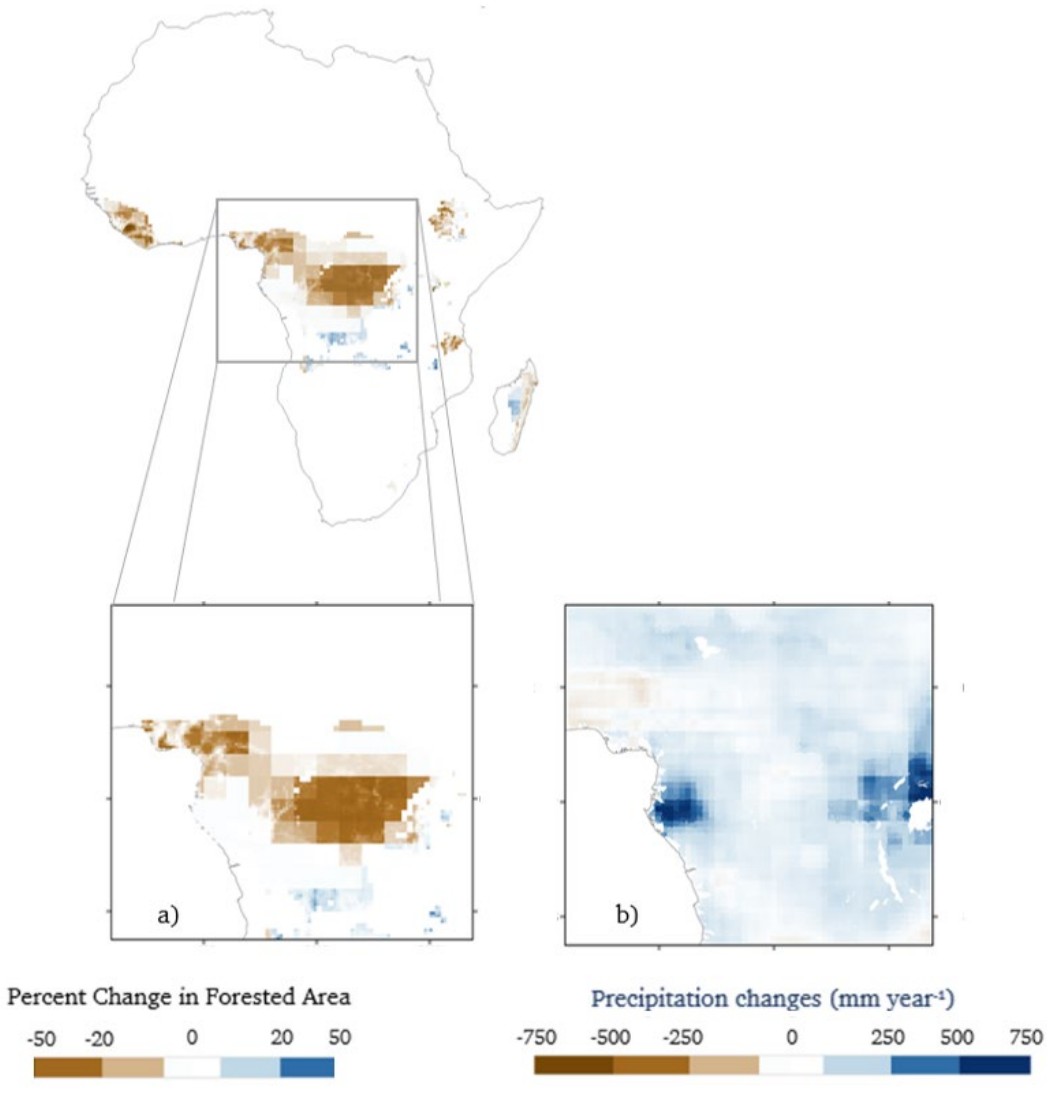

**460**   **Figure 20: (a) Percent Change in Forested Area under RCP 8.5. (b) Precipitation change under RCP 8.5 for the Central Africa. The changes are from the historical period (1975 - 2005) to the future period (2070 - 2100)**

The limitations of this modelling study include non-overlapping observation data, lack of reservoir management data, poor resolution input data, limited information on agricultural management practices, and an assumption about the source of irrigation water. These limitations are also mentioned in the study by Chawanda et al. (2020), but another limitation was

**465**   discovered for reservoirs that start working during the simulation period. These reservoirs did not let water pass through until



the year they became operational, which is erroneous. The bug was reported to the developers. There were also bugs in very large reservoirs where the reservoir continued to grow in surface area beyond the maximum surface area specified during model set up. This was specifically observed in Lake Nasser and few reservoirs in West Africa. We also aknowledte that the vegetation in our model does not respond to heat waves as it might in real world.

**5      Conclusion and recommendations**

In this study, we set up a SWAT+ model for the entire Africa using SWAT+ AW (Chawanda et al., 2020b). We implemented irrigation, reservoirs and HMBC. We also modified the model to incorporate transient land-use change and used the model to run projections under RCPs 2.6, 6.0 and 8.5 using 4 GCMs with and without land-use change.

We have demonstrated that apart from changes in precipitation, the response in ET plays a substantial role on whether water
availability will increase or decrease. The Niger River basin is likely to experience the largest reduction in the river flows under RCPs 6.0 and 8.5 while the Congo River basin is likely to experience reduced river flows under all RCPs. At the same time, the Limpopo will likely see an increase in river flows under future climate. The largest increases in river flows are observed in the Nile under RCPs 6.0 and 8.5.

The Congo basin is likely to have less ET and more water availability under combined effects of CC and LULCC. LULCC is
also likely to lead to more stress on water availability in the Senegal and the Orange River basins. The effects of LULCC on river flows point to the potential of using land use as an adaptation measure to future hydrological changes.

Further studies can be done based on the rich data produced in this study. While CC is likely to cause drier dry months and wetter wet months in most major river basins in Africa, it is still unclear whether the frequency, intensity, and duration of wet and dry extreme events will increase or decrease. Thus, further study is needed to analyse how climate change will affect the
frequency and duration of above-mentioned events. Further work needs to be done on the reservoirs in the SWAT+ code to allow water through a reservoir even before the year that the reservoir became operational. Multiple growing seasons should be implemented in the SWAT+ model as suggested by Nkwasa *et al.* (2020) based on available cropping patterns from global datasets to account for ET coming from rainfed and irrigated crops.

Results show that precipitation will be reduced mainly in North Africa, the Senegal region, and south-eastern Africa, including
Madagascar, while large increases are expected in the Ethiopian Highlands. ET will likely increase under CC around the equator region driven by an increase in precipitation in the same area, while North Africa, West Africa and most of Southern Africa will experience lower ET driven by a decline in average rainfall in those regions under CC. The changes in average river flows mostly depend on the change in water availability. Projections show that the Congo River basin is likely to experience lower average river flows in the future due to CC under all RCPs, while the Niger is likely to experience a strong
decrease in river flows under RCPs 6.0 and 8.5. The Limpopo (under all RCPs) and the Nile (under RCPs 6.0 and 8.5) are likely to experience higher average river flows in the future due to CC. The Nile, the Senegal, and the Orange have mixed



signals between RCPs, which calls for more careful planning of water resources for the future in these regions to account for the uncertainties in current-generation projections.

Combining CC and LULCC results in larger signals in changes in ET than only considering CC. The Congo basin has a strong
decrease in ET under LULCC and CC, yet it has an increase in ET under CC alone. The Congo River basin also experiences a shift from a decrease of river flows under pure CC scenarios to an increase in river flows when LULCC is considered. In the Senegal and the Orange River basins and Madagascar, increased agricultural activity negatively impacts water availability and increases ET.

The projected changes have huge implications on the livelihood of people in Africa. Increased rainfall and river flow variability
pose a threat of increased frequencies of floods and droughts and will likely threaten agricultural production across the continent. Africa is dominated by small-scale farming with farmers heavily dependent on rainfall (Thornton et al., 2014) which makes the livelihood of people on the continent vulnerable to projected hydrological changes. There is a need to establish agricultural policies and practices that increase resilience against CC. Hasan et al. (2019) estimate that a 10% decrease in future water resources would affect 57% of the African population for 2050 projections.

We have demonstrated that LULCC can have significant effects on water resources. Governments can adopt land use policies as one of the adaptation measures to counter the effects of CC. In addition, governments should develop land policies to stop the current deforestation trend on the continent as combined climate and land-use change can have a higher impact on water resources than pure CC. However, the impacts of pure CC should not be underestimated. CC has already had devastating impacts in many regions in Africa, from increased frequency of droughts to flooding. As such, policies that help curb
greenhouse gas emissions need to be employed all over the continent to limit CC.

**Funding:** The authors thank The Research Foundation – Flanders (FWO) for funding the International Coordination Action (ICA) "Open Water Network: Open Data and Software tools for water resources management" (Project Code G0E2621N) and the EU H2020 project Defining the future of inland water services for Copernicus (Water-ForCE), Grant agreement No 101004186.

**Data and code availability:** The tools used in this study are available from the HYDR repository (https://github.com/VUB-HYDR). Simulation results are in very big in size (Greater than 1 Terrabyte) and cannot easily be hosted online. These are available upon request. All input data is from open sources:

- The digital elevation model (DEM) data was taken from the Shutter Radar Topography Mission (Farr et al., 2007, downloadable from http://dwtkns.com/srtm).
- The land use map was prepared from the Land-Use Harmonization Project phase 2 (LUH2; Hurtt et al., 2011).
- Soil data with 250m× 250m resolution was obtained from the Africa Soil Information Service (AfSIS; Hengl et al., 2015).
- Monthly discharge observation data was obtained from Global Runoff Data Centre (GRDC, n.d.).



- Irrigated areas were obtained from Food and Agriculture Organisation at 0.083° × 0.083° resolution (FAO; Siebert & Frenken, 2014).

- Reservoir data was obtained from the Global Reservoir and Dam (GRanD) database (Beames et al., 2019).

- Yearly ET data was obtained from CSIRO's Moderate resolution imaging spectroradiometer reflectance Scaling Evapotranspiration (CMRSET; Guerschman et al., 2009) and Water Productivity through Open access of Remotely sensed derived data (WaPOR; FAO 2018) at the resolution of 0.0022° × 0.0022°.

- The EartH2Observe, WFDEI and ERA-Interim data Merged and Bias-corrected for ISIMIP (EWEMBI; Lange, 2016) dataset was obtained for weather forcing through the Inter-Sectoral Impact Model Intercomparison Project (ISIMIP) at a daily timestep. Additionally, daily weather forcing was obtained through ISIMIP for four bias-adjusted global climate models (GCMs) under the Representative Concentration Pathway (RCP) 2.6, 6.0, and 8.5: GFDL-ESM2M, HadGEM2-ES, IPSL-CM5A-LR, and MIROC5.

Please refer to Chawanda et al. (2020a) for a detailed description.

**Competing interests:** Ann van Griensven is a member of the editorial board of Hydrology and Earth System Sciences



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
