# Peer review of "Combined impacts of climate and land-use change on future water resources in Africa"

_Hydrology and Earth System Sciences, 2023_

## Author Comment (AC1)

**Author Responses to Reviewer Comments**

We would like to thank the reviewer for taking time to make comments and suggestions. We have revised the manuscript based on the feedback and have answered questions raised. We hope these revisions are satisfactory for the further processing of this paper.

In this response document, Orange is the quoted comment/question while dark blue is response text. *Italicised text is text extracted from the manuscript after implementing suggested changes.*

**Reviewer 1**

The authors used two experiments of CC and LULCC scenarios to evaluate their results. Yes, this is good to use but my concern is that what about the consideration of other factors that are interlinked to the oceans' atmospheric pressures as they may affect the precipitation distributions.

We thank the reviewer for raising this concern. In our study, the scenarios (Climate Change (CC) and Land Use and Land Cover Change (LULCC)) were informed by General Circulation Models (GCMs). These GCMs inherently account for interlinked processes that tie together oceanic conditions and atmospheric pressures, affecting precipitation distributions and even include those that pertain to ocean-atmosphere feedback loops, and the role of land use and land cover (Boucher et al., 2020; Flato et al., 2013; Mortier et al., 2020). In our study's context, these factors were considered as integral components of the underlying models, ensuring a comprehensive representation of precipitation distribution influenced by oceanic conditions.

The authors simulate the historical (1975 - 2005) and the far future (2070 - 2100) periods. I think it would have been better if they could produce the near (2021-2040) and midterm (2041-2070) results for better preparation of the societies for the climate variabilities that may evolve as the climate changes in the near future.

We appreciate the insightful comment raised here. We do acknowledge that near and midterm analyses have their merits, especially for operational planning and immediate adaptation strategies. Ideally, a comprehensive study would include both, but given the scope and focus of this study, we prioritized the end-of-century timeframe. The decisions made in areas such as infrastructure and land use planning often have consequences that last many decades. End-of-century projection data is crucial for these kinds of decisions.

In addition, some of the most significant impacts of climate change on catchments may not be readily apparent until later in the century. Our timeframe allows us to capture these longer-term transformations and understand the cumulative impacts of climate change and land use, which often have nonlinear effects that accelerate over time. Furthermore, focusing on the end of the century underscores the long-term consequences of current actions or inaction as emphasised a study by Thiery et al. (2021), reinforcing the ethical responsibility we have to future generations.

That said, we believe that our work can be complemented by other studies focusing on shorter-term impacts, thereby providing a full spectrum of data for decision-making.

Regarding model calibration, it is good that the authors have used NSE values. But some NSE values are still below the threshold values in some stations. Would have been added some other model performance measures would be good to crosscheck the results and see the outputs.

Large-scale models are often not adapted and evaluated at very large scales due to high computation time requirements or lack of information on human interactions, such as dam operations and irrigation practices at local scale (Chawanda et al., 2020). But these adaptations are essential for impact model assessment as demonstrated by Krysanova et al. (2018). To address this challenge the Hydrological Mass Balance Calibration (HMBC) (Chawanda et al., 2020) was developed. HMBC tunes the model to make sure the major hydrological processes such as Evapotranspiration and Surface Runoff better represent observations in the long term. Thus, aiming for a more accurate Hydrological Mass balance, all done with less computational requirements.

For these reasons, our study used HMBC and we did not expect very high NSE values because we did not calibrate routing parameters in the SWAT+ model setup (such as Channel Mannings Coefficient and Channel Hydraulic Conductivity, among others). Calibrating for specific river flows would require more runs estimated in the thousands (and may be in the tens of thousands considering many gauging stations considered). For such a large-scale model, this would be impractical, hence, HMBC.

However, by looking at the river flows we could see how the improvements in the model's representation of internal processes improved river runoff simulation even though we did not calibrate for it. We have clarified this in section 4.1 as follows:

> *The model was calibrated using HMBC following Chawanda et al. (2020). This methodology aims to match long term averages of major components of the hydrological cycle. However, even though HMBC does not calibrate against river flows, looking at the changes in performance of river flows may reflect any improvements (if any) in the representation of internal processes.*

Figure 2: I couldn't see figures 1a and 1b, Could you please provide and level it in the figure. And the same is in Figure 3 as well.

Thanks for catching that. This has been done.

In the results section authors have repeatedly used the latitude grids to spatially elaborate their results. That is good but it could have been better if at least one map of Africa could show the latitude and longitude values so that we can easily understand where it would be they are referring to.

This feedback has been incorporated. Figure 1 now includes coordinates.

[Figure]

Figure 1: Major river basins in Africa

In section 3.1.2, it could be good if the authors explain how they evaluate the model performance in quantifying the ET performance across the river basins.

We thank the reviewer for this comment. We realise this element was indeed missing. We have adapted the manuscript in Section 3.1.2 to reflect that the comparisons between WaPOR and SWAT+ ET was based on spatially averaged annual ET per major basin:

> *Further ET checks were done by comparing simulated and WaPOR spatially averaged annual ET values for each major basin. The model captured the low ET values expected in the Sahara, Namib and Kalahari Deserts (Figure 5 a and b). However, the SWAT+ model overestimated ET in the Congo Basin by 58 mm year$^{-1}$ and underestimated annual ET in the lower Nile River by 42 mm year$^{-1}$. The high observed ET values (locally up to 500 mm year$^{-1}$) in the lower Nile, which the model underestimates, are expected due to irrigation activity and multiple cropping sessions in the area.*

**References**

Boucher, O., Servonnat, J., Albright, A. L., Aumont, O., Balkanski, Y., Bastrikov, V., Bekki, S., Bonnet, R., Bony, S., Bopp, L., Braconnot, P., Brockmann, P., Cadule, P., Caubel, A., Cheruy, F., Codron, F., Cozic, A., Cugnet, D., D'Andrea, F., … Vuichard, N. (2020). Presentation and Evaluation of the IPSL-CM6A-LR Climate Model. *Journal of Advances in Modeling Earth Systems*, *12*(7). https://doi.org/10.1029/2019MS002010

Chawanda, C. J., Arnold, J., Thiery, W., & van Griensven, A. (2020). Mass balance calibration and reservoir representations for large-scale hydrological impact studies using SWAT+. *Climatic Change*, *163*(3), 1307–1327. https://doi.org/10.1007/s10584-020-02924-x

Flato, G., Marotzke, J., Abiodun, B., Braconnot, P., Chou, S., & Collins, W. (2013). Evaluation of climate models. *Climate Change 2013 the Physical Science Basis: Working Group I Contribution to the Fifth Assessment Report of the Intergovernmental Panel on Climate Change*, *9781107057*, 741–866.

Krysanova, V., Donnelly, C., Gelfan, A., Gerten, D., Arheimer, B., Hattermann, F., & Kundzewicz, Z. W. (2018). How the performance of hydrological models relates to credibility of projections under climate change. *Hydrological Sciences Journal*, *63*(5), 696–720. https://doi.org/10.1080/02626667.2018.1446214

Mortier, A., Gliß, J., Schulz, M., Aas, W., Andrews, E., Bian, H., Chin, M., Ginoux, P., Hand, J., Holben, B., Zhang, H., Kipling, Z., Kirkevåg, A., Laj, P., Lurton, T., Myhre, G., Neubauer, D., Olivié, D., von Salzen, K., … Tilmes, S. (2020). Evaluation of climate model aerosol trends with ground-based observations over the last 2~decades -- an AeroCom and CMIP6 analysis. *Atmospheric Chemistry and Physics*, *20*(21), 13355–13378. https://doi.org/10.5194/acp-20-13355-2020

Thiery, W., Lange, S., Rogelj, J., Schleussner, C.-F., Gudmundsson, L., Seneviratne, S. I., Andrijevic, M., Frieler, K., Emanuel, K., Geiger, T., Bresch, D. N., Zhao, F., Willner, S. N., Büchner, M., Volkholz, J., Bauer, N., Chang, J., Ciais, P., Dury, M., … Wada, Y. (2021). Intergenerational inequities in exposure to climate extremes. *Science*, *374*(6564), 158–160. https://doi.org/10.1126/science.abi7339

---

## Author Comment (AC2)

**Author Responses to Reviewer Comments**

We would like to thank the reviewer for taking time to make comments and suggestions. We have revised the manuscript based on the feedback and have answered questions raised. We hope these revisions are satisfactory for the further processing of this paper.

In this response document, Orange is the quoted comment/question while blue is response text. *Italicised text is text extracted from the manuscript after implementing suggested changes.*

**Reviewer 2**

The model calibration part has been completely referred to a previous paper. It is necessary to explain the relevant calibration principles and details here in brief. Please try to make the paper self explanatory to the extent possible. It can be provided as appendix if the authors feel so. The term soft data has been defined only in Chawanda et al. (2020a) and not here.

We thank the reviewer for raising this comment. We have adopted the suggestion and included a paragraph in Section 2.3 describing Hydrological Mass Balance Calibration (HMBC) and included a definition of Soft Data referencing Arnold et al. (2015).

> *Unlike traditional calibration methods which predominantly rely on hard data, such as time series of hydrological quantities at a specific point in the watershed, the Hydrological Mass Balance Calibration (HMBC) uses soft data to improve model accuracy, especially for larger scale applications. Soft data refers to information on individual processes, such as long-term annual average estimates (Arnold et al., 2015). This type of data provides insights into the broader patterns and averages, setting constraints during hard calibration to enhance the representation of hydrological processes. Using soft data reduces computational and time expenses (Chawanda et al 2020a). HMBC aims to adjust model parameters to ensure that the simulated long-term average water balance components align with observed averages which enhances the model's performance in impact studies by more accurately simulating hydrological mass balance components. The procedure involves running the model, evaluating results against soft data, estimating new parameter values, re-running the model, and repeating this cycle until certain criteria are met. Generally, a hydrological component such as ET is calibrated within five iterations before progressing to the next component in each region.*

We have also revised more areas to make the manuscript stand more independent where brief overviews were missing.

I could not read the previous paper in full (Chawanda et al., 2020a). However, I think that this paper is somewhat similar to the previous paper referred here in terms of assessment of climate change impact. A comparison of results over the common landmass is needed to be presented in the current manuscript.

Chawanda et al, (2020a) introduces HMBC and demonstrates the role of conducting calibration on Climate Change (CC) projections made. However, as CC assessment itself was not the focus of the paper, there are significant differences that prevent a direct comparison between the results from that paper and the results presented in this study. To begin with, the previous paper's projections were made only using RCP 6.5 and for a different period 2060 - 2090, not 2070 – 2100. This is in addition to having different historical reference period (1970 – 2000 versus 1976 – 2005 for the current study) For these reasons, a one-to-one comparison could not be made. However, it is worth noting that the signal for change in ET and Runoff for CC RCP 6.0 is consistent despite these period differences.

What is the benefit of doing this climate change and LULC study? The results are averaged over large basins for long time periods. Hence, how beneficial this study will be for local scale adaptation or management?

Addressing the broader implications and concerns of climate change and LULC (Land Use and Land Cover) is indeed essential at both the large and local scales. While our report offers results at the major-basin level to provide an encompassing perspective, the depth and granularity of our model simulations and data inputs are usable at national and regional levels for more local-scale insights and adaptation strategies. We based our model on a 300m resolution Digital Elevation Model to ensure that local topographical variations are represented. Our challenge was with the land use harmonization project (LUH2) data (Hurtt et al., 2020) which is presented as netCDF. This data format is not compatible with SWAT+ and the resolution was 0.25 decimal degrees. However, since each pixel had the percent cover for each land use type contained in the pixel, details were not really lost for simulation purposes (Figure 1).

[Figure]

*Figure 1: Contrast between the raster format accepted by SWAT+ and the Format of Land Use data from the Land Use Harmonisation Project 2 (LUH2)*

We adapted SWAT+ to use this data and we describe the details in Chawanda et al. (2020a). Furthermore, the applicability and utility of our model at intra-basin level are underscored by its successful use in two separate studies focusing on sediments (Nkwasa et al., 2022a) and crop management representations (Nkwasa et al., 2022b), both within the Nile basin. Thus, indeed we present holistic basin-level views while retaining the potential for local insights as can also be seen in the maps.

Results should be presented for near term periods (e.g. 2030-2050, 2050-2070). Only presenting long term results may not be useful and verifiable in a possible time frame.

We thank the reviewer for the valuable feedback. We concur that presenting findings for nearer-term periods, such as 2030-2050 and 2050-2070, holds is valuable especially when considering strategies and immediate responses to climatic and shifts and land use changes.

However, the emphasis of our study was on end-of-century projections. This is primarily due to the lasting implications decisions in areas like infrastructure development and land-use management carry, often spanning several decades. By focusing on end-of-century data, we highlight the extended consequences these decisions might have, making it instrumental for far-reaching planning. Moreover, many of the profound repercussions of climate change on catchments might only emerge more overtly towards the latter part of the century. Our chosen timeline allows us to identify the evolving transformations and better understand the cumulative effects of both climate change and land use changes. Thus, highlighting projections toward the century's end serves as a reminder of the lasting implications of our current choices—or lack thereof—as argued in studies like that of Thiery et al. (2021). Such a perspective underlines our ethical commitments to the generations that follow.

Nevertheless, we fully endorse the idea that more studies that zero in on more immediate impacts and in specific regions would greatly complement this study to better guide and inform on both near and far term periods.

It has been mentioned that over some zones where streamflow data is not available, calibration was done only using ET. Highlight those zones and explain how this ET calibration was done.

Notably, we mention Congo and Nile, with the latter experiencing significant restrictions in river flow data access. In these specific regions, the HMBC was constrained to just ET data, thus initial parameters that are derived from inputs are used to estimate surface runoff and groundwater components, but ET parameters were optimised to align simulations with remotely sensed ET datasets for these areas. We refer to supplementary material, which includes a map highlighting areas within major basins where only ET was used in the HMBC process.

*In contrast to a previous Southern Africa SWAT+ model application* (Chawanda et al., 2020a)*, the Nile and Congo River basins had very few gauging stations from which long term average surface runoff could be derived. This was a major problem in the Nile Basin where river data availability from public sources is even more restricted. As such, only ET was calibrated by HMBC in some calibration zones where gage data was not available (Figure S1).*

[Figure]

Figure S1: A highlight of areas within major river basins where Hydrological Mass balance (HMBC) only used Evapotranspiration (ET) Soft Data.

The calibration is based on preserving the long term averages of the water cycle components. Please explain conceptually how this can be used to model yearly dynamics of the water cycle.

Thanks for this comment. SWAT+ models these dynamics based on inputs such as climate data and parameters derived from soil maps, land use maps and topography data from DEM. This includes seasonal cycles of river flow, ET and other variables. By calibrating for long-term averages, we ensure that internal processes are better represented in the long-term in addition to the seasonal variations already simulated by the default model configuration. However, to better capture more detailed temporal river flow patterns instead of long-term, a more detailed calibration approach is required which was beyond the scope of this study.

I understand that ground water component for calibration was calculated as a residual of longterm averaged water budget. Hence, the uncertainties in the other datasets would have propagated to the GW data. Please explain how far this will affect the result.

We appreciate the reviewer's insight on this aspect of our hydrological modeling study. It is indeed a correct assessment that when calculating the groundwater component as a residual of the long-term averaged water budget, uncertainties from other datasets may propagate to the calculated groundwater data. This is indeed a limitation when direct observational data for a component like groundwater is lacking, especially over large scales like Africa.

We would like to highlight a few points regarding our approach. Firstly, at the scale of our study, direct observations of groundwater are virtually nonexistent, making it challenging to calibrate or validate groundwater components using traditional methods. Secondly, while there are uncertainties associated with each component of the water balance, the relative impact of these uncertainties on the groundwater component, when computed as a residual, is not straightforward since some uncertainties might offset each other, while others might accumulate. However, propagating the uncertainties to ground water was beyond the scope of this study.

While we acknowledge the potential for uncertainties to propagate to the calculated ground water data, our approach was a pragmatic solution given the data constraints. We are keen to refine our methodologies as more granular and accurate data becomes available.

Is it safe to assume that the catchment proeprties and the model parameters are stationary in time for such a long time period? How to account for the non-stationarity in catchment properties and model parameters?

The reviewer brings an important issue of non-stationarity in catchment properties and model parameters over long periods. As correctly pointed out earlier, our projections incorporate Land Use and Land Cover (LULC) changes, which account for a significant portion of the variability in catchment properties by considering variations due to factors such as urbanisation, deforestation, agricultural activities, and much more.

In terms of topography, it's generally safe to assume that significant changes do not occur over the typical timeframes of hydrological studies, unless there are extreme events like large-scale landslides or significant human-made modifications. For the duration of our study, we believe this assumption holds. However, while the fundamental soil type remains relatively constant over time, properties such as soil compaction or organic matter content, can change due to land management practices and natural processes. Thus, it is indeed important to note that in specific catchments where intensive land-use activities occur, there might be some level of change in soil properties.

As per the question to address non-stationarity in catchment properties, in studies where fundamental changes to topography or soil are observed in each catchment, one could consider adaptive calibration techniques where the model is recalibrated at regular intervals to adjust to changing conditions. In this case Dynamic Parameters would be useful to allow for parameters to change over time based on predefined rules or relationships. This can capture some of the non-stationarity inherent in catchments. At present, SWAT+ limits such updates to the curve number parameter through decision tables (Arnold et al., 2018).

Please use continous colour bar instead of ranges for presenting ET, rainfall and their differences (figures 5-9). The class ranges are quite large and a continous colour bar will provide more information that these maps I feel. Especially, with the current ET difference maps, I get an impression that there are large differences between the SWAT+ ET and WaPOR ET. A continous colour bar may help to understand this better.

Thanks for this suggestion. We have adopted it in the revised manuscript.

Please limit the y-axis value range in figure 10. I think the flow value is maximising at 40000 cubic metre per second. This will help us visualise the temporal variation in the flow better.

This has been done.

Hydrological modelling exercises are generally not useful for simulating extreme events such as floods and droughts. Please include your views on how to model extremes under future climate change and land cover change scenarios.

Thanks for raising such an important issue about the utility of hydrological modeling in capturing extreme events. While hydrological modeling has its challenges, especially in simulating extremes, it remains a valuable tool when used appropriately.

Modern hydrological models are able to capture both average and extreme conditions which studies have demonstrated (Peredo et al., 2022; van Kempen et al., 2021). The effectiveness of a hydrological model in simulating extreme events depends on the quality, extent and objectives of the calibration and validation process and whether datasets that include extreme events are used (Onyutha, 2019). In addition, incorporating projections from climate models and scenarios of land cover change in simulations allows hydrological models to simulate potential changes in hydrological extremes under different future scenarios. An example: if a climate model projects more intense rainfall events in the future, a hydrological model can simulate the resulting flood conditions given it accounts for land cover change scenarios. Downscaling techniques can also play a role in simulating hydrological extremes as they allow high-resolution meteorological inputs from coarse-resolution climate models which makes a difference in areas with high climate variability in space (such as mountainous areas).

Some hydrological models have more detailed representation of physical processes, allowing them to better simulate extreme conditions. For example, SWAT+ has been coupled with GW-Flow and modflow ground water modules which makes it better at simulating droughts. Thus, we argue otherwise, that hydrological models are very essential and useful in simulating extremes.

**References**

Arnold, J. G., Bieger, K., White, M. J., Srinivasan, R., Dunbar, J. A., & Allen, P. M. (2018). Use of decision tables to simulate management in SWAT+. *Water (Switzerland)*, *10*(6), 1–10. https://doi.org/10.3390/w10060713

Arnold, J. G., Youssef, M. A., Yen, H., White, M. J., Sheshukov, A. Y., Sadeghi, A. M., Moriasi, D. N., Steiner, J. L., Amatya, D. M., Skaggs, R. W., Haney, E. B., Jeong, J., Arabi, M., & Gowda, P. H. (2015). Hydrological processes and model representation: Impact of soft data on calibration. *Transactions of the ASABE*, *58*(6), 1637–1660. https://doi.org/10.13031/trans.58.10726

Chawanda, C. J., Arnold, J., Thiery, W., & van Griensven, A. (2020a). Mass balance calibration and reservoir representations for large-scale hydrological impact studies using SWAT+. *Climatic Change*, *163*(3), 1307–1327. https://doi.org/10.1007/s10584-020-02924-x

Hurtt, G. C., Chini, L., Sahajpal, R., Frolking, S., Bodirsky, B. L., Calvin, K., Doelman, J. C., Fisk, J., Fujimori, S., Klein Goldewijk, K., Hasegawa, T., Havlik, P., Heinimann, A., Humpenöder, F., Jungclaus, J., Kaplan, J. O., Kennedy, J., Krisztin, T., Lawrence, D., … Zhang, X. (2020). Harmonization of global land use change and management for the period 850–2100 (LUH2) for CMIP6. *Geoscientific Model Development*, *13*(11), 5425–5464. https://doi.org/10.5194/gmd-13-5425-2020

Nkwasa, A., Chawanda, C. J., Jägermeyr, J., & Van Griensven, A. (2022b). Improved representation of agricultural land use and crop management for large-scale hydrological impact simulation in Africa using SWAT+. *Hydrology and Earth System Sciences*, *26*(1), 71–89. https://doi.org/10.5194/HESS-26-71-2022

Nkwasa, A., Chawanda, C. J., & Van Griensven, A. (2022a). Regionalization of the SWAT+ model for projecting climate change impacts on sediment yield: An application in the Nile basin. *Journal of Hydrology: Regional Studies*, *42*. https://doi.org/10.1016/j.ejrh.2022.101152

Onyutha, C. (2019). Hydrological Model Supported by a Step-Wise Calibration against Sub-Flows and Validation of Extreme Flow Events. *Water*, *11*(2), 244. https://doi.org/10.3390/w11020244

Peredo, D., Ramos, M.-H., Andréassian, V., & Oudin, L. (2022). Investigating hydrological model versatility to simulate extreme flood events. *Hydrological Sciences Journal*, *67*(4), 628–645. https://doi.org/10.1080/02626667.2022.2030864

Thiery, W., Lange, S., Rogelj, J., Schleussner, C.-F., Gudmundsson, L., Seneviratne, S. I., Andrijevic, M., Frieler, K., Emanuel, K., Geiger, T., Bresch, D. N., Zhao, F., Willner, S. N., Büchner, M., Volkholz, J., Bauer, N., Chang, J., Ciais, P., Dury, M., … Wada, Y. (2021). Intergenerational inequities in exposure to climate extremes. *Science*, *374*(6564), 158–160. https://doi.org/10.1126/science.abi7339

van Kempen, G., van der Wiel, K., & Melsen, L. A. (2021). The impact of hydrological model structure on the simulation of extreme runoff events. *Natural Hazards and Earth System Sciences*, *21*(3), 961–976. https://doi.org/10.5194/nhess-21-961-2021